# One Life to Learn: Inferring Symbolic World Models for Stochastic Environments from Unguided Exploration

**Zaid Khan**[1]     **Archiki Prasad**[1]     **Elias Stengel-Eskin**[2]     **Jaemin Cho**[3]     **Mohit Bansal**[1]
[1]UNC Chapel Hill     [2]UT Austin     [3]Johns Hopkins
{zaidkhan, archiki, mbansal}@cs.unc.edu, esteng@utexas.edu, jaemin@jhu.edu

onelife-worldmodel.github.io

## ABSTRACT

Symbolic world modeling is the task of representing an environment's transition dynamics as an executable program. Previous research on symbolic world modeling has focused on simple, deterministic environments with abundant data and human-provided guidance. We treat symbolic world modeling as a problem of autonomous scientific discovery in a complex, dangerous, and stochastic environment. An agent has "one life" to explore the environment, and must infer the dynamics that govern it from observation alone, without external guidance in the form of environment-specific rewards, goals, or prior knowledge. Our framework, ONELIFE, decomposes the transition function into a mixture of small, probabilistic laws, each with a precondition that determines when it applies and an effect that governs a narrow slice of the state. Laws activate only when relevant, forming a computation graph specific to each transition. Gradients flow only through active laws, giving precise credit assignment over a large, hierarchical state space. To support evaluation, we contribute Crafter-OO, a reimplementation of Crafter that exposes a structured, object-oriented state and a pure transition function, together with an evaluation suite of 30+ scenarios and evaluation protocols for measuring (a) state ranking, the ability to distinguish plausible future states from implausible ones, and (b) state fidelity, the ability to generate future states that closely resemble reality. ONELIFE learns key dynamics from a single unguided episode, outperforming a strong baseline on 16 of 23 scenarios tested. We also demonstrate the world model's utility for planning, where rollouts simulated within the world model successfully identify superior strategies in multi-step goal-oriented tasks. Our work establishes a foundation for autonomously constructing programmatic world models of unknown, complex environments.

## 1 INTRODUCTION

World modeling is a critical task in artificial intelligence, providing an agent with a functional understanding of its environment's underlying dynamics. By learning a world model, an agent can predict the outcomes of its actions without having to actually interact with the real world. One line of research in world modeling aims to learn symbolic world models via program synthesis (i.e., representing worlds models with code) with a view towards building representations that are interpretable, editable, and verifiable by humans.

While such approaches have been successful in environments with a limited number of discoverable mechanics and low stochasticity (Piriyakulkij et al., 2025; Tang et al., 2024; Dainese et al., 2024) these assumptions are often violated in more complex environments. Examples of such environments are popular open-world sandbox games (e.g. MineCraft, RuneScape) containing numerous, diverse mechanics spanning crafting, combat, and physics. These more realistic environments have irreducible stochasticity (e.g., outcomes of actions are subject to random chance), a lack of extrinsic rewards (e.g., players set their own goals and there is no well-defined criteria for "winning"), and

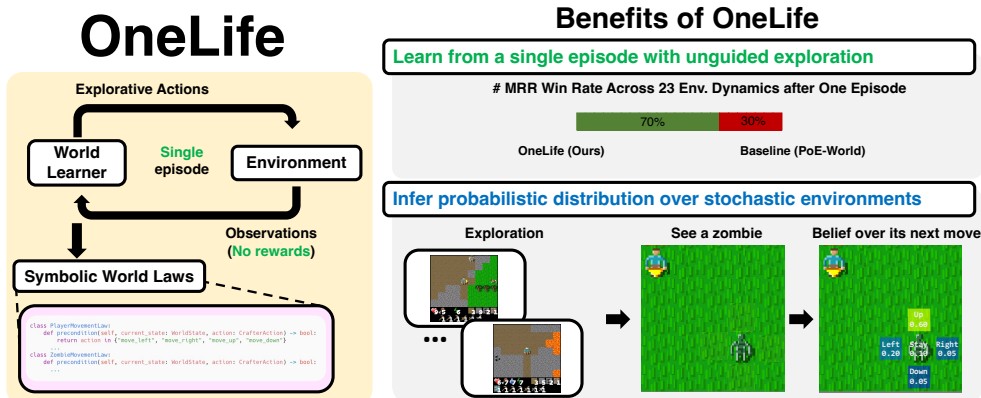

Figure 1: ONELIFE synthesizes world laws from a single unguided (no environment-specific rewards / goals) episode in a hostile, stochastic environment. ONELIFE models the world as mixture of laws in code with a precondition-effect structure, each governing an aspect of the world, and infers parameters for the mixture that best explain the observed dynamics of the world. The resulting world model (WM) provides a probability distribution over attributes of an object-oriented world state, such as the `position` of a particular zombie. ONELIFE outperforms a strong baseline in modeling 16/23 core game mechanics tested, measured by MRR (Mean Reciprocal Rank) of the true next state (Sec. 4) under the WM's likelihood. See Box B.3 for a synthesized zombie law.

a high cost of exploration (e.g., entering dangerous areas without preparation can result in death), making it crucial to learn from minimal interaction. This leads to our central research question:

> *How can an agent reverse engineer the laws of a complex, dangerous stochastic world, given a limited interaction budget and without environment-specific human-specified goals or rewards?*

We introduce a framework for symbolic world modeling, ONELIFE, a name that reflects our focus on learning a symbolic world model from a single episode with unguided exploration. As illustrated in Fig. 1 (top-right), ONELIFE learns from just a *single, unguided run* in the environment, a contrast to previous work (Piriyakulkij et al., 2025; Tang et al., 2024; Dainese et al., 2024) that assumes access to a large number of interactions as well as environment specific guidance provided by humans (e.g., goals / rewards designed for the environment). ONELIFE recovers a **program** that describes the environment's underlying transition dynamics $p(s_{t+1}|s_t, a_t)$ which models the probability distribution $p$ over next states $s_{t+1}$ given a current state $s_t$ and action $a_t$. The agent performs this inference using only observations, **without access to rewards or other domain-specific guidance.** ONELIFE has two key components: **a law synthesizer** (Sec. 3.3) that proposes new laws and **an inference algorithm** (Sec. 3.4) that re-weights laws based on their predictive ability over observations. Crucially, the inference algorithm is gradient-based and only updates the laws that alter the observed variables between current state $s_t$ and predicted next state $s_{t+1}$, allowing for efficient and targeted learning. These components work together in a probabilistic programming mixture-of-laws approach (Sec. 3.2) that proposes and re-weights rules based on whether the preconditions for the laws to be applicable are met and the effect of the predictions w.r.t. the observed environment transitions. This approach enables our model to infer distributions over complex, stochastic events, as shown in the Fig. 1 (bottom-right), where a learned world model outputs a distribution over a zombie's next move. Crucially, ONELIFE not only produces a distribution over states but *learns from* stochastic observations; the true movement of the zombie in Fig. 1 also follows a distribution, which ONELIFE seeks to approximate.

To evaluate our approach, we first created a suitable testbed – Crafter-OO – by re-engineering the complex Crafter (Hafner, 2022) environment to be a pure function $T(s, a) \rightarrow s'$ of a structured, text-based hierarchical object-oriented world state. In other words, all the information needed to compute the next state is represented in a single structured, object-oriented representation, and there is a ground-truth program for the transition function that computes the next environment state purely from the state representation, without any "hidden variables". This text-based, object-oriented representation is natively readable by LLMs and thus allows them to try reconstructing the transition function by writing code that programatically modifies the structured state. It allows for a structured,

object-oriented representation that is directly comprehensible to language models and enables symbolic reasoning over a world with rich entity interactions. We introduce a new evaluation protocol that uses two axes (Sec. 4): **state ranking**, the ability to distinguish valid outcomes from invalid ones according to the world's laws, and **state fidelity**, the ability to produce plausible future states for planning. Our experiments show that ONELIFE better captures the environment's dynamics compared to several baselines, including PoE-World (Piriyakulkij et al., 2025), showing improved ability to simulate future states given a state and candidate action, and to distinguish between likely and unlikely outcomes of an action. We further show that the learned model supports planning in imagination; by simulating rollouts of different policies entirely within the model, we can evaluate and distinguish between effective and ineffective strategies for multi-step goal-oriented tasks.

In summary, our contributions include:

- ONELIFE, a probabilistic symbolic world model that can learn from stochastic and hostile environments with minimal interactions and without access to human-defined rewards. ONELIFE outperforms prior work, learning a world model that better predicts true environment dynamics.
- Crafter-OO, a reimplementation of Crafter (Hafner, 2022) that exposes a structured, object-oriented symbolic state and and a pure transition function that operates on that state alone. This enables us to test ONELIFE in a complex, stochastic environment and lays the groundwork for future work in symbolic world modeling and programmatic reinforcement learning.
- An evaluation suite for world modeling within Crafter / Crafter-OO with 30+ executable scenarios that test knowledge of all core mechanics in Crafter and a pool of mutators that can programatically generate illegal distractor states to probe world model understanding alongside, new state fidelity and state ranking metrics for evaluating world models in complex, stochastic, environments.

## 2 RELATED WORK

**Symbolic World Models.** Symbolic world models represent an environment's transition dynamics as executable code, producing interpretable, editable, and generalizable models from limited data. Prior work has used LLMs to synthesize a single, monolithic program that functions as a world model (Tang et al., 2024; Dainese et al., 2024). Piriyakulkij et al. (2025) introduced a compositional approach by representing the world model as a product of programmatic experts, enabling modeling of more complex dynamics. Other methods have synthesized programs for planning (Ahmed et al., 2025) or combined functional and automata synthesis to capture latent state dynamics (Das et al., 2023). LLMs have also been used to construct formal planning representations like PDDL from environment interactions or text for symbolic planners (Guan et al., 2023; Deng et al., 2024). Our work differs from these methods in three aspects. First, we operate in a complex, open-world environment based on Crafter (Hafner, 2022) with stochasticity and many interacting mechanics, whereas prior work has operated in simpler, often deterministic domains (e.g., grid-worlds or Atari games). Second, we do not assume abundant interaction data: our agent learns from a limited budget obtained in a single episode – or life. Third, ONELIFE learns without external rewards or human-specified goals, framing the task as unguided reverse engineering of the environment's laws.

**Programmatic Representations for Decision-Making.** Program synthesis has been used to represent other components of intelligent agents. Programmatic policies have been shown to offer greater interpretability and generalization compared to neural networks (Trivedi et al., 2021; Liang et al., 2022). LLMs have been used to generate programmatic reward functions from natural language instructions, enabling agents to pursue complex, user-specified objectives (Ma et al., 2024; Yu et al., 2023; Klissarov et al., 2025). Programs have been used to build libraries of composable, temporally extended skills, allowing agents to solve long-horizon tasks by combining previously learned behaviors (Wang et al., 2025; Stengel-Eskin et al., 2024). These methods focus on representing components of the agent's internal decision-making process: *how it should act* (policies), *what it should value* (rewards), or *what it is capable of doing* (skills). In contrast, our work learns a model of *how the external world behaves*; this task-agnostic model of environment dynamics is complementary to policies, rewards, and skills, and supports planning and decision-making for any downstream goals.

**World Modeling for Open-Ended Exploration and Discovery.** Agents that explore and learn in complex, open-world environments without extrinsic rewards typically learn non-symbolic, latent world models and use them to drive exploration through intrinsic motivation (Hafner et al., 2023;

Micheli et al., 2023; Dedieu et al., 2025; Schwarzer et al., 2021). These agents plan using their world models to find novelty or surprise in their environments, discovering useful skills without task-specific supervision (Sekar et al., 2020). This connects to automated scientific discovery, which requires autonomously forming hypotheses and performing experiments to understand unknown systems (Jansen et al., 2024; Chen et al., 2025; Geng et al., 2025). New evaluation frameworks have been proposed to assess an agent's ability to rapidly induce world models in novel contexts (Ying et al., 2025; Vafa et al., 2024). Unlike methods that learn implicit, latent world models, our work learns an explicit, symbolic representation of the world's laws. We frame learning as reverse engineering a complex system's rules from unguided, limited interaction.

**Relation to Domain Inference and State Tracking.** ONELIFE tackles the challenge of *domain inference* (learning transition dynamics) rather than *state tracking* (inferring state from partial observations) (Gordon et al., 1993). Classical domain inference methods (Cresswell et al., 2009; Zhuo & Kambhampati, 2013) often rely on deterministic PDDL representations. We use a neurosymbolic, Python-based formalism because (1) standard PDDL cannot easily capture the stochastic dynamics of Crafter-OO, and (2) LLMs are much better at generating Python than Probabilistic PDDL.

## 3 OVERVIEW OF ONELIFE

Our framework, **ONELIFE** is designed to learn symbolic world models from a single, unguided episode of exploration. It is built on two key abstractions, a programmatic representation of world dynamics as a mixture of modular *laws* with learnable weights and an *observable extractor* that decouples the environment's state from the learning process. The framework consists: a **a world model as a program** (Sec. 3.2), a **law synthesizer** that proposes new laws using offline data from an **unguided exploration policy** (Sec. 3.3), an **inference algorithm** that re-weights laws based on observations (Sec. 3.4), and a **forward simulation process** that uses the learned model for predicting future states (Sec. 3.5).

We model the environment as having a pure, but potentially stochastic, transition function $T : \mathcal{S} \times \mathcal{A} \to \Delta(\mathcal{S})$, where $\Delta(\mathcal{S})$ is the space of probability distributions over the state space $\mathcal{S}$. This functional view aligns with modern reinforcement learning environment frameworks (Freeman et al., 2021; Matthews et al., 2024) and physical models, where the future state of a system is a pure function of an explicit state and any interventions. (See Section H for a discussion on why we model a pure transition function probabilistically.)

### 3.1 CRAFTER-OO: A TESTBED FOR SYMBOLIC WORLD MODELING

A common design assumption in previous work on symbolic world modeling (Tang et al., 2024; Piriyakulkij et al., 2025; Dainese et al., 2024) is that we have access to an object-oriented world state to use as input to the symbolic world model under construction. In practice, this state is only easily accessible for simple environments such as Minigrid (Chevalier-Boisvert et al., 2023) or BabyAI (Chevalier-Boisvert et al., 2018). Programmatic access to the state of more complex environments such as Atari games as used by Piriyakulkij et al. (2025) is only possible due to standalone development efforts such as OCAtari (Delfosse et al., 2024) which makes the internal object-oriented state of these environments accessible to researchers. The lack of an environment with an exposed, object-oriented state that is more complex than gridworlds or with mechanics more diverse than Atari games has thus far prevented evaluation and development of symbolic world modeling approaches for more complex environments. To close this gap, we implement Crafter-OO (Sec. C), which emulates the Crafter (Hafner, 2022) environment and action space (Tab. 3) by operating purely on an explicit, object-oriented game state [1] (Listing 1). Additionally, we contribute utilities for programmatically modifying the game state to create evaluation scenarios (Sec. E, Sec. 4.1).

Our target environment Crafter-OO features significant stochasticity, diverse forms of mechanics, and active non-player characters. This includes elements such as hostile and friendly agents with diverse, inherently random behaviors. Our framework is designed to infer the rules governing these in-

---

[1] We describe the state in Python/JSON because we found it substantially easier for LLMs to manipulate than PDDL. PDDL representations of our complex state became prohibitively large, increasing experimental costs. Furthermore, while Probabilistic PDDL can capture stochastic dynamics, it makes synthesis significantly more difficult, likely because Probabilistic PDDL is much rarer in pre-training data than standard PDDL or Python.

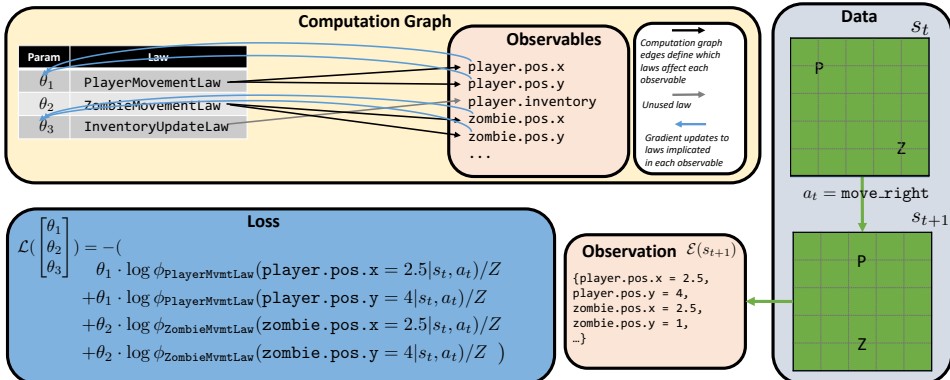

Figure 2: Illustration of the inference process. The active laws for each observable (defined by $\mathcal{I}_k(s_t, a)$) determine the structure of the computation graph, i.e., which laws and their corresponding parameters $\theta_i$ are related to which observables. This structure in turn informs the parameter updates. Shown here is a dataset with a single transition instance, in which the player (P) moves right; at the same time, a zombie (Z) independently moves left. this implicates two laws, `PlayerMovementLaw` and `ZombieMovementLaw`, while not implicating the `InventoryUpdateLaw`. As a result, the loss computation is only a function of $\theta_1$ and $\theta_2$. Note we use $Z$ here to denote the normalizing factor. Examples of synthesized laws can be seen in Sec. B.

teractions from observation alone, without access to rewards or human-specified goals. For instance, in Fig. 2, the scenario contains a "zombie" character chasing the player via stochastic movements. While one cannot perfectly predict the future position of a zombie due to inherent randomness built into the environment, our world model is able to capture this "chasing the player" behavior without any explicit supervision by predicting a discrete distribution for the `zombie.position` attributes.

## 3.2 ONELIFE: WORLD MODEL AS A MIXTURE OF LAWS

We consider environments with complex, structured state spaces $\mathcal{S}$ where the full state $s \in \mathcal{S}$ may be hierarchical and contain a mixture of entity types and attributes. An agent interacts with the environment by taking an action $a \in \mathcal{A}$ and observing a transition from state $s_t$ to $s_{t+1}$, as illustrated in Fig. 2. We model an environment's transition function as a composition of programmatic laws. A law, $L_i$, is a program defined by a pair $(c_i, e_i)$, where $c_i(s, a) \rightarrow \{\text{true}, \text{false}\}$ is a *precondition* and $e_i(s, a) \rightarrow \Delta(\mathcal{S})$ is an *effect*. The precondition determines whether the law is applicable to a state-action pair $(s, a)$. The effect function specifies a probability distribution over next states by defining distributions over the values of state attributes. For example, the `PlayerMovementLaw` in Fig. 2 applies to state-action pairs with a player and a move action, and has an effect on the player position's $(x)$ observable. This precondition-effect structure is inspired by classical planning and provides a natural way to specify the scope of each law, ensuring modularity (McDermott et al., 1998). During any given transition, multiple or no laws may be applicable.

To create a tractable interface to compare states predicted by a world model and the true state of the environment, we introduce an **observable extractor**, $\mathcal{E} : \mathcal{S} \rightarrow \mathcal{O}$. This function maps a complex state $s$ into a vector of primitive-valued **observables** $o \in \mathcal{O}$. In the scenario sketched in Fig. 2, the next state $s_{t+1}$ can be complex, with additional entities and objects (e.g., trees, inventory items, etc.). Nevertheless, one can tractably compare states via observations, i.e., *changes* between $s_t$ and $s_{t+1}$ such as `player.position`, `player.inventory`, `zombie.position`, etc. Note that any given law $L_i$ only makes predictions about a subset of all possible observables. For instance, in Fig. 2, the `PlayerMovementLaw` *only* makes predictions about `player.position` observables and *does not predict* the `zombie.position` observables.

Our world model can be viewed as a probabilistic program (van de Meent et al., 2021) that generates the next state's observables $o'$ conditioned on the current state $s$ and action $a$. The set of laws $\{L_i\}$ defines the components of this program. Formally, the effect $e_i(s, a)$ yields a set of conditional probability distributions $\{\phi_{i,o}\}_{o \in \mathcal{O}}$, where $\phi_{i,o}(o = v | s, a)$ is the distribution over possible values $v$ for observable $o \in \mathcal{O}$, with $v \in \text{supp}(o)$ denoting a specific outcome in the discrete support of the observable. Our implementation currently covers categorical and discrete distributions. In principle,

the framework extends to continuous distributions, as the inference algorithm only requires the ability to query the likelihood of an observed data point. For a given state-action pair $(s, a)$, the set of *active laws* is $\mathcal{I}(s, a) = \{i \mid c_i(s, a) \text{ is true}\}$ (e.g., `PlayerMovementLaw` and `ZombieMovementLaw` in Fig. 2). The model assumes that all observables are *conditionally independent* given the current state and action. The predictive distribution for a single observable $o$ is formed by combining the predictions from all active laws that have an opinion on it. Let $\mathcal{I}_o(s, a) = \{i \in \mathcal{I}(s, a) \mid o \in \mathcal{O}\}$ be the set of active laws relevant to observable $o$. The probability of observing an outcome $v$ for this observable is given by a weighted-product of conditional probability distribution from each law, parameterized by $\boldsymbol{\theta}$:

$$p(o = v|s, a; \boldsymbol{\theta}) \propto \prod_{i \in \mathcal{I}_o(s,a)} \phi_{i,o}(o = v|s, a)^{\theta_i} \tag{1}$$

The complete predictive distribution over the next state $s'$ is the product of the individual observable distributions:

$$p(s'|s, a; \boldsymbol{\theta}) = \prod_{o \in \mathcal{O}} p(o|s, a; \boldsymbol{\theta}) \tag{2}$$

The learnable weights $\boldsymbol{\theta}$ perform model selection over the set of candidate laws. Because the synthesizer generates a large pool of atomic laws, including incorrect hypotheses, the optimization process drives the weights of invalid laws toward zero to remove them from the model. Additionally, the weights enable multiple plausible laws to vote on the final predictive distribution, allowing the model to aggregate conflicting predictions.

**Comparison to Prior Product-of-Expert Formalisms.** Although we use a product-of-experts structure like PoE-World (Piriyakulkij et al., 2025), the underlying representation and optimization differ fundamentally. Conceptually, PoE-World learns a superposition of experts where each program predicts the *entire* next state. This causes the posterior to be noisy in complex environments, as irrelevant experts contribute uniform predictions to attributes they do not model well. In contrast, ONELIFE factorizes the transition function into *atomic laws* that individually predict a *minimal subset* of the next state (e.g., only the player's health, or only a specific map tile). This atomicity enables our optimization procedure (Sec. 3.4) to construct a *dynamic computational graph* for every transition. By exploiting the precondition-effect structure, we route gradients only to laws relevant to the specific observed transition. This avoids the "static graph" limitation of prior work and allows ONELIFE to scale to diverse object-oriented attributes (e.g., inventory items, map tiles, NPC states) beyond the simple physics variables (e.g., position, velocity) targeted by prior work.

### 3.3 ONELIFE: UNGUIDED ENVIRONMENT EXPLORATION AND LAW SYNTHESIS

The set of candidate laws $L_i$ is generated from unguided agent-environment interactions through a two-stage process. First, an autonomous exploration policy gathers a corpus of interaction data. Second, a synthesizer proposes candidate laws that explain the state transitions observed in this data.

**Exploration Policy.** Previous work in symbolic world modeling often assumes access to curated offline datasets or utilizes online interaction guided by human-provided goals or environment rewards. In our unsupervised setting, such guidance is *unavailable*. Furthermore, in a hostile environment such as Crafter-OO, a simple random policy fails to survive long enough to experience the diverse mechanics necessary for comprehensive world modeling. Therefore, we employ an exploration policy driven by a large language model. The policy is not provided with specific knowledge of the environment; instead, it is given the high-level objective to discover as many underlying mechanics as possible, treating exploration as a reverse-engineering task. We distinguish between general genre priors and environment-specific dynamics. General genre priors are high-level concepts common to the class of open-world survival environments, such as the existence of hostile entities, the ability to collect resources, or the ability to craft tools. In contrast, environment-specific dynamics refer to exact rules, such as "Zombies chase players" or "Wood is needed to make a pickaxe." The exploration policy is provided with the former to prevent aimless behavior typical of random policies, but it is strictly withheld from the latter. This mimics a realistic scenario where an agent enters a new environment possessing broad intuition about the genre, but must reverse-engineer the specific laws and mechanics of that unique world from scratch. We use the agent scaffolding from Balrog (Paglieri et al., 2025) to implement the agent. The agent's architecture maintains a rolling window of its recent state-action history to provide context for decisions. The prompt (see Sec. G)

also instructs the agent to maintain a transient summary of its current understanding of the world's rules, refining its hypotheses as it interacts with the environment.

**Law Synthesizer.** The synthesizer is an automated routine that queries a Large Language Model (LLM) to explain observed state transitions. Our system operates by iterating through every transition in the exploration dataset and performing a systematic comparison of the object-oriented state at each timestep. This process automatically identifies the specific object attributes that have changed, such as an entity's position or an inventory count, without requiring manual specification of what to track. For each identified change, the routine queries the LLM to output a Python class containing `precondition` and `effect` methods. The `effect` method is generated to explicitly perform the observed attribute assignment on a state object. This process yields *atomic* laws that govern minimal subsets of state attributes. For instance, a complex combat event is automatically decomposed into separate candidate laws where one explains the health decrease and another explains the enemy's movement. This modularity allows the subsequent inference stage (Sec. 3.4) to perform precise credit assignment by isolating correct mechanics from incorrect hypotheses. We provide examples of synthesized laws in Sec. B.

**Synthesis Differences From PoE-World.** While Piriyakulkij et al. (2025) adopt a specialized approach utilizing a bank of over 30 synthesizers equipped with prompts tailored to pre-identified mechanics, ONELIFE employs a single synthesizer. This setup *requires* our agent to identify mechanics on the fly and codify them without prior knowledge of the environment's rules. Consequently, our synthesizer consumes the entire game state in a general-purpose format (JSON) to write code for diverse aspects of the world, including map tiles, entities, and player inventories, whereas PoE-World limited synthesis to a specific set of physics-based attributes.

## 3.4 ONELIFE: INFERENCE ON LAW PARAMETERS

We learn the weight vector $\boldsymbol{\theta}$ by maximizing the log-likelihood of a dataset of observed transitions $\mathcal{D} = \{(s_t, a_t, s_{t+1})\}_{t=1}^N$. For clarity, we first define the loss for a single transition $(s, a, s')$; the total loss is the sum over all transitions in the dataset.

Based on the conditional independence of observables, the negative log-likelihood for a single transition decomposes into a sum over each observable $o \in \mathcal{O}$:

$$\mathcal{L}(\boldsymbol{\theta}; s, a, s') = -\sum_{o \in \mathcal{O}} \log p(v_o^*|s, a; \boldsymbol{\theta}) \tag{3}$$

where $v_o^* = \mathcal{E}(s')_o$ is the ground truth value of observable $o$ extracted from the next state $s'$. The log-probability term is derived from the combined predictions of the active laws. Let $\mathcal{I}_o(s, a)$ be the set of active laws that make a prediction for observable $o$. We first define the combined, unnormalized log-score for any potential value $v$ as the weighted sum of log-scores from these laws. The weights $\theta_i$ are the *only learnable parameters*:

$$\ell_o(v|s, a; \boldsymbol{\theta}) = \sum_{i \in \mathcal{I}_o(s,a)} \theta_i \cdot \log \phi_{i,o}(o = v|s, a) \tag{4}$$

Normalized log-probability of observing the specific outcome $v_o^*$ is then given by the log-softmax function. Let $\text{supp}(o)$ be the discrete support (set of all possible values) for observable $o$:

$$\log p(v_o^*|s, a; \boldsymbol{\theta}) = \ell_o(v_o^*|s, a; \boldsymbol{\theta}) - \log \sum_{v \in \text{supp}(o)} \exp\left(\ell_o(v|s, a; \boldsymbol{\theta})\right) \tag{5}$$

The optimization process leverages the dynamic computation graph induced by our law structure. For each transition and each observable, the loss gradient is calculated with respect to the weights $\theta_i$ only for the active laws $i \in \mathcal{I}_o(s_t, a_t)$. This effectively **routes** credit for an outcome exclusively to the laws that made a prediction about it. This sparse, targeted update mechanism provides more precise credit assignment than methods that update a global set of weights based on aggregate outcomes. We use L-BFGS for optimization (Nocedal & Wright, 2006).

## 3.5 ONELIFE: FORWARD SIMULATION AND LIKELIHOOD

Forward simulation is the process of using the learned world model generatively to predict a future state $\hat{s}_{t+1}$ given a current state $s_t$ and an action $a_t$. By generating rollouts of future trajectories,

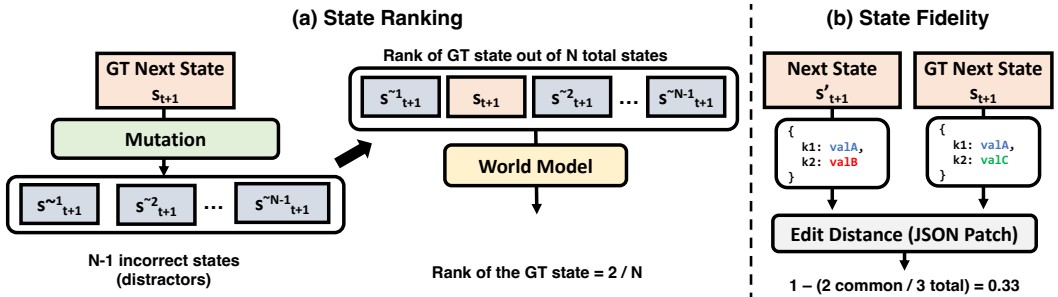

Figure 3: Two evaluation metric categories described in Sec. 4. A world state of an environment usually has more than two keys (*i.e. Crafter-OO's state (Section C.2) when populated has 100+ key-value pairs,*) and often has nested values, but here we show a simplest case to explain the calculation of (normalized) edit distance. We create distractors for state ranking using *mutators* (Sec. D), which programatically modify the next state $s'$ in a transition $(s, a, s')$ to be illegal under the true transition function. For example, one of our mutators allows a crafting action (e.g. making a stone pickaxe) to succeed even when the prequisites for the crafting are not met.

an agent can evaluate action sequences against a specific goal or reward function without costly or irreversible real-world interaction.

The simulation of a single timestep from $(s_t, a_t)$ involves a multi-step sampling and reconstruction process. First, for each observable $o \in \mathcal{O}$, the model forms a predictive probability distribution $p(o|s_t, a_t; \boldsymbol{\theta})$. This distribution is constructed by identifying the set of active laws $\mathcal{I}_o(s_t, a_t)$ relevant to that observable and combining their predictions according to their learned weights $\theta_i$, as specified in Equation 1. This distribution can be used to evaluate the likelihood of an observable conditioned on $(s, a)$ pair. Second, a concrete outcome $\hat{v}_o$ can be sampled from this distribution for each observable: $\hat{v}_o \sim p(o|s_t, a_t; \boldsymbol{\theta})$. This the collection of sampled outcomes $\{\hat{v}_o\}_{o \in \mathcal{O}}$ is used to construct the full symbolic next state $\hat{s}_{t+1}$. A reconstruction function, which mirrors the observable extraction process, assembles these values back into the environment's structured state representation.

## 4 EVALUATION PROTOCOLS AND METRICS

The evaluation of world models for a stochastic environment is non-trivial. An useful world model fulfills two criteria: (a) **state ranking**, the ability to distinguish plausible future states from implausible ones, and (b) **state fidelity**, the ability to generate future states that closely resemble reality. Both are illustrated in Fig. 3.

**State Ranking (Fig. 3 (a)).** These metrics assess the model's ability to rank the true next state higher than the distractors. To create the distractor states, we use **mutators**, which are programmatic functions that apply semantically meaningful, rule-breaking changes to the true next state. For example, a mutator could change a character's position to a location they cannot physically reach. We include details on mutators in Sec. D.

- **Rank @ 1 (R@1):** A binary metric that measures whether the model correctly assigns the highest probability (rank 1) to the true next state among all candidates.
- **Mean Reciprocal Rank (MRR):** This metric averages the reciprocal rank of the correct answer across all test instances. A higher MRR indicates that the model consistently ranks the correct state higher. The formula is: MRR $= \frac{1}{N} \sum_{i=1}^{N} \frac{1}{r_i}$, where $r_i$ is the rank of the ground truth state for the $i$-th transition, with rank 1 being the highest probability. We favor MRR over raw mean rank because its inverse scaling $(1/r)$ heavily penalizes missing the top rank, reflecting the high cost of sampling invalid states during planning. Furthermore, unlike mean rank, MRR provides a standardized score invariant to the candidate set size $N$, which varies in our setup depending on the number of applicable mutators.

**State Fidelity (Fig. 3 (b)).** These measure the error between predicted and ground truth states.

Table 1: Performance comparison of world modeling methods on the Crafter-OO environment, averaged over ten trials. We evaluate models on two criteria: **state fidelity** and **state ranking** All methods use the ONELIFE exploration policy and law synthesizer but differ in their parameter inference method. ONELIFE shows significant improvements over the PoE-World inference algorithm and ONELIFE variant without parameter inference. The random baseline is shaded in  gray . The "No Inference" row ablates learnable law parameters $\theta$ from our world model.

| Law Synthesis | Law Param. Inference | State Ranking | | State Fidelity | |
| --- | --- | --- | --- | --- | --- |
| (Sec. 3.3) | (Sec. 3.4) | Rank @ 1 ↑ | MRR ↑ | Raw Edit Dist. ↓ | Norm. Edit Dist. ↓ |
| Random World Model | | 8.5% | 0.322 | 121.538 | 0.809 |
| WorldCoder | | 0.0% | 0.264 | 27.180 | 0.181 |
| ONELIFE | PoE-World | 10.8% | 0.351 | 10.634 | 0.071 |
| ONELIFE | No Inference | 13.0% | 0.429 | 8.540 | 0.057 |
| ONELIFE | ONELIFE | 18.7% | 0.479 | 8.764 | 0.058 |
| Δ over PoE-World | | (+7.9%) | (+0.128) | (-1.870) | (-0.013) |

- **Raw Edit Distance:** The total number of atomic JSON Patch operations required to transform the predicted state, $s'_{t+1}$, into the ground truth state, $s_{t+1}$.
- **Normalized Edit Distance:** The raw edit distance divided by the total number of elements in the state representation.

## 4.1 EVALUATION FRAMEWORK IMPLEMENTATION ON CRAFTER-OO

Evaluating a world model on random rollouts may not provide sufficient coverage of rare or important events in an environment. To ensure our evaluation is comprehensive, we create evaluation trajectories from a suite of **scenarios**. Each scenario runs short, scripted policy from an initial state designed to reliably exercise a specific game mechanic or achieve a particular goal, ensuring that our evaluation thoroughly covers the environment's dynamics. We generate a comprehensive evaluation dataset by implementing scenarios that cover every achievement in the game's achievement tree, seen in Fig. 4. This ranges from basic actions like collecting wood to complex, multi-step tasks like crafting an iron sword, ensuring all of the game's core mechanics are tested. More details on scenarios are provided in Sec. E. We generate distractors for each transition in the evaluation dataset using a bank of mutators which each produce a subtle, but illegal transformation of the game state in response to an action. Some examples are causing an incorrect item to be produced when taking a crafting action, or allowing an item to be produced without the correct requirements, or illegal entity behavior such as teleporting. Because mutators have specific preconditions (e.g., combat mutators only apply during combat), the number of applicable distractors varies per state. In our experiments, the total candidate set size (ground truth plus distractors) ranges from $N = 7$ to $N = 11$ per transition. Details on mutators and the evaluation are provided in Sec. D and Sec. F.

## 5 EXPERIMENTAL SETUP AND RESULTS

We conduct a series of experiments to evaluate ONELIFE. First, we quantitatively assess the model's predictive accuracy using our state ranking and fidelity metrics across a comprehensive suite of scenarios. Second, we test the model's ability to support planning in imagination. We use the model to perform simulated rollouts of different policies, evaluating whether it can predict the outcomes of these plans well enough to distinguish effective strategies from ineffective ones (Sec. A).

We compare ONELIFE against three baselines (fully detailed in Section I): a **Random World Model**; **PoE-World** (Piriyakulkij et al., 2025), the prior state-of-the-art symbolic framework that learns a product of experts; and **WorldCoder** (Tang et al., 2024), which differs from PoE-World by synthesizing a monolithic, deterministic program for the transition function.

## 5.1 RESULTS

**State Fidelity and Ranking.** ONELIFE learns a world model with significantly higher predictive judgment than baseline methods while maintaining competitive generative fidelity. Table 1 compares our full method against baselines and key ablations across all evaluation metrics. ONELIFE's primary advantage appears in the predictive judgment metrics. We achieve a discriminative accuracy

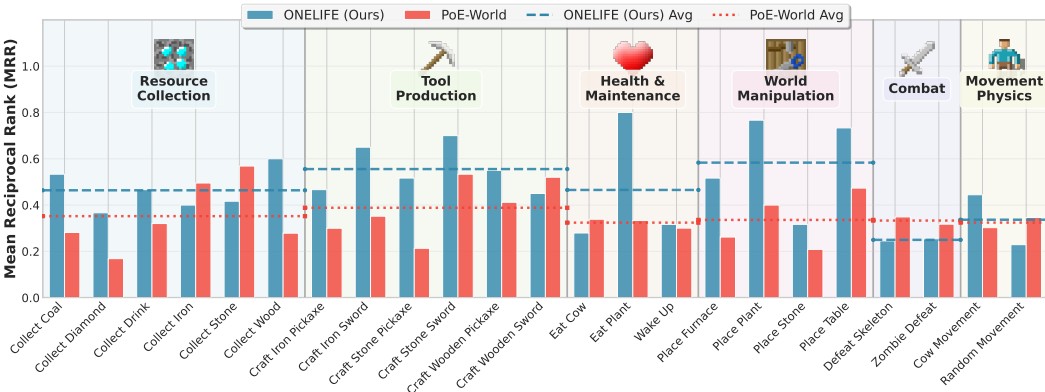

Figure 4: Per-scenario state ranking performance of ONELIFE (Ours) versus PoE-World, measured by Mean Reciprocal Rank (MRR ↑). Scenarios are grouped by the core game mechanic they test. Horizontal lines show the average MRR across all scenarios in a group for ONELIFE and PoE-World. ONELIFE demonstrates a more accurate understanding of the environment's laws, achieving a higher average MRR and outperforming the baseline on the majority of individual scenarios.

of 18.7% and an MRR of 0.479, outperforming the PoE-World optimization baseline by 7.9 percentage points and 0.128, respectively. While precisely generating a complex future state remains challenging, our model has learned an accurate understanding of the environment's underlying laws. This enables it to assign high probability to valid transitions and low probability to invalid ones. The comparison to the "random world model" shows that (i) a high edit distance can quickly be amassed if the world models updates observables that are unchanged in the ground truth state, thus, reinforcing why such simulation is challenging; (ii) optimizing for generative metrics like state fidelity alone *does not* yield a better world model to guide an agent, e.g., while the PoE-world model (row 2 in Tab. 1) dramatically improves the state fidelity by reducing the edit distance *a factor of 10*, it only *marginally* improves the ability to *rank multiple states* by ≈2% over random (Rank@1) – reiterating the need for state ranking metrics. Removing the parameter inference step ("ONELIFE & None") results in a performance drop of 5.7% in Rank@1 and 0.05 in MRR, confirming that the weights are essential for distinguishing valid laws from incorrect ones.

**Fine-grained Evaluation.** Figure 4 breaks down Mean Reciprocal Rank performance across individual scenarios spanning mechanics from resource collection to combat. ONELIFE consistently outperforms the PoE-World baseline on the majority (16/23) of scenarios. These improvements stem from a robust understanding of the environment's diverse rules rather than strong performance on only a few simple mechanics.

## 6 CONCLUSION

We address the problem of learning a symbolic world model from limited, unguided interaction in a complex, stochastic environment. We introduced ONELIFE, a framework that represents world dynamics as a probabilistic mixture of modular, programmatic laws. Its core learning mechanism routes credit for observed state changes exclusively to the laws responsible for predicting them, enabling effective learning even when many rules are inactive during a given transition. Evaluated on Crafter-OO, our variant of the complex Crafter environment with object-centric state, ONELIFE learns a world model with superior predictive judgment compared to a strong baseline, more accurately distinguishing plausible future states from implausible ones. This improvement is consistent across a wide range of game mechanics. Our work provides a foundation for building agents that can autonomously reverse engineer the rules of an unknown environment.

## ACKNOWLEDGEMENTS

This work was supported by NSF-AI Engage Institute DRL-2112635, ARO Award W911NF2110220, ONR Grant N00014-23-1-2356, DARPA ECOLE Program No. HR00112390060, Capital One Research Award, Apple PhD Fellowship, and NDSEG PhD

Fellowship. The views contained in this article are those of the authors and not of the funding agency.

## ETHICS STATEMENT

We do not foresee any ethical implications beyond standard ethical and safety considerations that apply to AI research generally.

## REPRODUCIBILITY STATEMENT

We have open-sourced Crafter-OO, ONELIFE, and the evaluation framework used in our work to aid reproducibility. All prompts and key details of the exploration policy, synthesis algorithm, and law parameter inference have been described.

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

## A  PLANNING AND MULTI-STEP SIMULATION WITH THE LEARNED WORLD MODEL

To assess the practical utility of the learned world model, we evaluate its effectiveness in a planning context. Our protocol tests the model's ability to distinguish between effective and ineffective plans through forward simulation. For a set of scenarios, we define a reward function and two distinct, programmatic policies (plans) to achieve a goal within the scenario. Each plan is represented as a hierarchical policy (in code) that composes subroutines for navigation, interaction, and crafting.

We give an example in box G.3 for the "Zombie Fighter" scenario. Each reward function is likewise written in code and calculates rewards from the rollout of a plan. We execute rollouts of both plans within our learned world model and, separately, within the ground-truth environment. The measure of success is whether the world model's simulation yields the same preference ranking over the two plans as the true environment, based on the final reward. This assesses if the model has captured the causal dynamics necessary for goal-directed reasoning.

**Setup.** We design three scenarios that test distinct aspects of the environment's mechanics: combat, tool-use and resource consumption, as shown in Table 2. In the **Zombie Fighter** scenario, an agent with low health must defeat two zombies. The superior plan involves a multi-step process: pathfinding to locate and harvest trees, crafting a table and then a sword, and only then engaging in combat. The alternative is to fight immediately. The **Stone Miner** scenario tests the model's understanding of resource collection. The effective plan is to first harvest wood, craft a pickaxe, pathfind to a stone, and then mine. Attempting to mine stone directly is ineffective. Finally, the **Sword Maker** scenario evaluates knowledge of resource consumption. The goal is to craft multiple swords. The efficient plan places a single crafting table and reuses it, whereas the inefficient plan wastes wood by placing a new table for each sword. On average, a plan requires $\approx 18$ steps to execute, with the longest plans

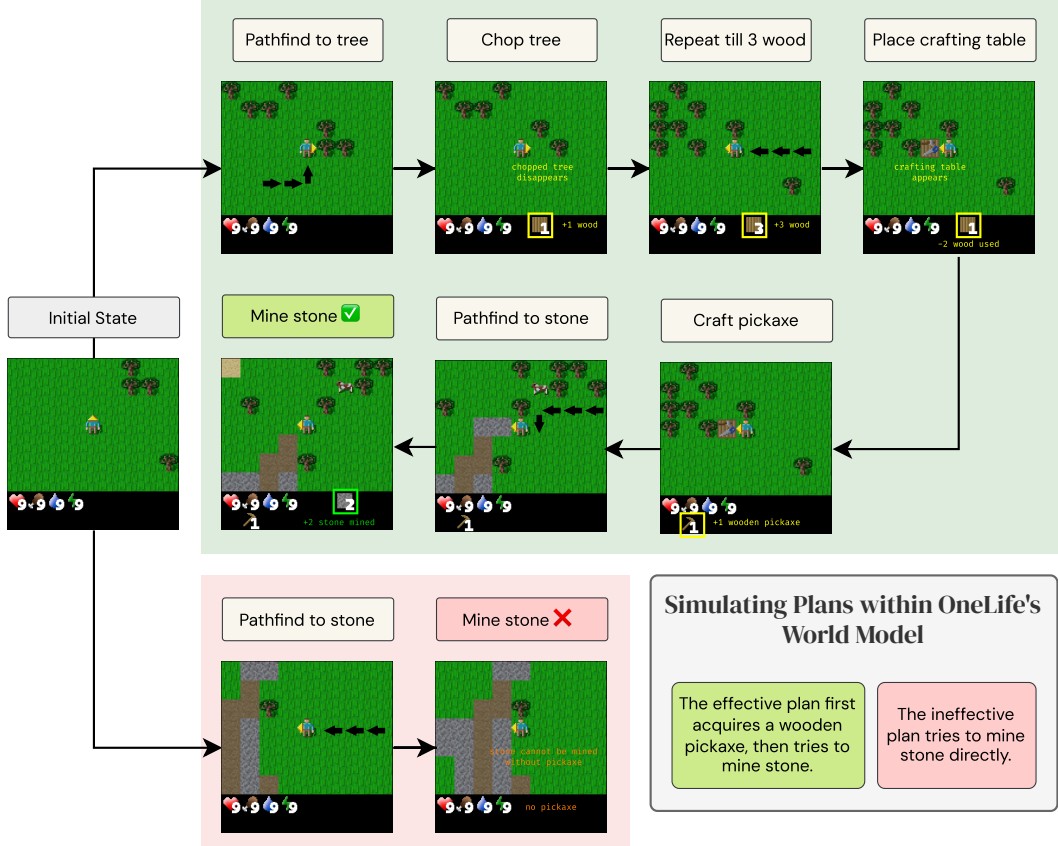

Figure 5: We show an example of plan execution *within* ONELIFE's world model for the "Stone Miner" scenario. The task is to mine stone, and can only be successfully completed if a wooden pickaxe is obtained before attempting to mine stone. We simulate two plans within the world model. The effective plan carries out a multi-step sequence of gathering wood, crafting a wooden pickaxe, and then attempting to mine. The ineffective plan attempts to mine the stone directly. The world learned by ONELIFE correctly simulates causal game mechanics that cause the effective plan to succeed and the ineffective plan to fail. The frames are generated by rendering the structured states constructed by ONELIFE's learned transition function.

taking $> 30$ steps. Thus, simulating the results of these plans tests the ability of the world model to accurately model the consequences of long sequences of actions upon the world. We show an example of plan execution in imagination for the "Stone Miner" scenario in Fig. 5.

**Results.** Table 2 shows that across all three scenarios, our learned world model correctly predicts the more effective plan. The ranking of plans generated by simulating rollouts in ONELIFE matches the ranking from the ground-truth environment. For instance, in the Zombie Fighter scenario, the model correctly simulates that the multi-step plan of crafting a sword leads to higher Damage Per Second, identifying it as the superior strategy. This demonstrates that ONELIFE captures a sufficiently accurate causal model of the world to support basic, goal-oriented planning.

# B   LAW EXAMPLES

Below, we give examples of various laws synthesized by ONELIFE. In box B.1 and box B.2, we show examples of how ONELIFE has learned the hierarchical structure of Crafter-OO/Crafter's tech-tree. In this case, one must mine stone before a stone pickaxe can be produced. These laws are deterministic in the sense that they define probability distributions that place mass 1.0 on a single outcome, consistent with the probabilistic framework defined in Sec. 3.2. In box B.3, we give

Table 2: Planning via forward simulation. Our learned world model is used to compare alternative plans in three scenarios. This is done by executing the plans in the world model, and measuring the reward obtained by each plan. In each case, ONELIFE produces the same ranking over plans as the ground-truth environment, demonstrating its ability to capture causally relevant dynamics for goal-directed decision-making and accurately simulate long action sequences of $> 30$ steps. Each plan was executed 10 times.

| Scenario | Plan Description | Reward Function | Avg. Steps | True Env. | | ONELIFE's WM | |
|---|---|---|---|---|---|---|---|
| | | | | Reward | Preferred | Reward | Preferred |
| Zombie Fighter | Harvest Wood → Craft Table → Craft Sword → Fight | Damage Per Second | 33 | 2.0 | ✓ | 2.03 | ✓ |
| | Fight Immediately | | 17 | 1.0 | | 1.67 | |
| Stone Miner | Harvest Wood → Craft Table → Craft Pickaxe → Mine | Stone Collected | 31 | 3.0 | ✓ | 3.0 | ✓ |
| | Mine Immediately | | 13 | 0.0 | | 0.0 | |
| Sword Maker | Reuse Crafting Table for all Swords | Swords Crafted | 5 | 4.0 | ✓ | 4.0 | ✓ |
| | Place New Table per Sword | | 10 | 2.0 | | 2.0 | |

an example of a law synthesized by ONELIFE for a stochastic mechanic, in this case, the chase behavior of zombies when they are within a certain range of a player. The idle skeleton law in box B.6 and moving skeleton law in box B.4 make conflicting predictions; these are aggregated by the weight inference process in Sec. 3.4 to produce a distributional prediction that takes into account the predictive accuracy of both laws.

---

**Box  B.1 | Mine Stone Law**

```python
class MineStoneLaw:
    def __init__(self):
        """Initialize with configurable parameters."""
        pass

    def precondition(self, current_state: WorldState, action: str) -> bool:
        """Return True if this law should apply to the given state and action."""
        if action != "Do":
            return False

        target_material, _ = current_state.get_target_tile()

        if target_material == "stone":
            # Check if the player has any pickaxe
            has_pickaxe = (
                current_state.player.inventory.wood_pickaxe > 0 or
                current_state.player.inventory.stone_pickaxe > 0 or
                current_state.player.inventory.iron_pickaxe > 0
            )
            return has_pickaxe

        return False

    def effect(self, current_state: WorldState, action: str) -> None:
        """Apply the law by modifying the world state."""
        # Increment stone in inventory
        current_state.player.inventory.stone = DiscreteDistribution(
            support=[current_state.player.inventory.stone + 1]
        )

        # Replace the mined stone material with grass
        current_state.set_facing_material("grass")
```

Box B.2| Craft Stone Pickaxe

```python
class CraftStonePickaxe:
    def __init__(self):
        """Initialize with configurable parameters."""
        # No specific parameters needed for this crafting recipe.
        pass

    def precondition(self, current_state: WorldState, action: str) -> bool:
        """Return True if this law should apply to the given state and action."""
        # Check if the action is "Make Stone Pickaxe"
        if action == "Make Stone Pickaxe":
            # Check if player has required materials
            has_wood = current_state.player.inventory.wood >= 1
            has_stone = current_state.player.inventory.stone >= 1
            return has_wood and has_stone
        return False

    def effect(self, current_state: WorldState, action: str) -> None:
        """Apply the law by modifying the world state."""
        # Decrease wood by 1
        current_state.player.inventory.wood = DiscreteDistribution(support=[
        current_state.player.inventory.wood - 1])
        # Decrease stone by 1
        current_state.player.inventory.stone = DiscreteDistribution(support=[
        current_state.player.inventory.stone - 1])
        # Increase stone_pickaxe by 1
        current_state.player.inventory.stone_pickaxe = DiscreteDistribution(
        support=[current_state.player.inventory.stone_pickaxe + 1])
```

Box B.3| Zombie Chase

```python
class ZombieAggroMovement:
    def __init__(self):
        """Initialize with configurable parameters."""
        pass # No specific parameters are needed for this observed law.

    def precondition(self, current_state: WorldState, action: str) -> bool:
        """Return True if this law should apply to the given state and action."""
        # This law applies if there are any ZombieState entities within the
        player's
        # update range, as their movement is an autonomous process.
        zombies_in_range = current_state.get_object_of_type_in_update_range(
        ZombieState)
        return len(zombies_in_range) > 0

    def effect(self, current_state: WorldState, action: str) -> None:
        """Apply the law by modifying the world state."""
        player_pos = current_state.player.position

        # Retrieve all ZombieState objects that are within the update range.
        # This implicitly filters for zombies close enough to be active/
        observable.
        zombies_to_update = current_state.get_object_of_type_in_update_range(
        ZombieState)

        for zombie in zombies_to_update:
            # Calculate the differences in coordinates between the player and the
         zombie.
            dx = player_pos.x - zombie.position.x
```

```
24              dy = player_pos.y - zombie.position.y
25
26              # Initialize new positions to current positions (no movement by
        default)
27              new_x = zombie.position.x
28              new_y = zombie.position.y
29
30              # Prioritize movement along the X-axis
31              if dx != 0:
32                  # Move one step towards the player along the X-axis.
33                  new_x = zombie.position.x + (1 if dx > 0 else -1)
34              elif dy != 0:
35                  # If X-axis is already aligned, move one step towards the player
        along the Y-axis.
36                  new_y = zombie.position.y + (1 if dy > 0 else -1)
37
38              # Update the zombie's position in the state using
        DiscreteDistribution.
39              zombie.position.x = DiscreteDistribution(support=[new_x])
40              zombie.position.y = DiscreteDistribution(support=[new_y])
```

### Box B.4| Skeleton Movement

```
1  class SkeletonRandomMovementLaw:
2      def __init__(self):
3          """Initialize with configurable parameters."""
4          pass
5
6      def precondition(self, current_state: WorldState, action: str) -> bool:
7          """Return True if this law should apply to the given state and action."""
8          # This law applies generally to all skeletons, independent of player
        action for movement
9          return True
10
11     def effect(self, current_state: WorldState, action: str) -> None:
12         """Apply the law by modifying the world state."""
13         skeletons = [obj for obj in current_state.objects if isinstance(obj,
        SkeletonState)]
14
15         for skeleton in skeletons:
16             current_x = skeleton.position.x
17             current_y = skeleton.position.y
18
19             # Possible next X positions: current_x, current_x + 1, current_x - 1
20             skeleton.position.x = DiscreteDistribution(support=[
21                 current_x,
22                 current_x + 1,
23                 current_x - 1
24             ])
25             # Possible next Y positions: current_y, current_y + 1, current_y - 1
26             skeleton.position.y = DiscreteDistribution(support=[
27                 current_y,
28                 current_y + 1,
29                 current_y - 1
30             ])
```

**Box B.5| Health Regeneration Law**

```python
class PlayerInventoryHealthRegeneration:
    def __init__(self, max_health: int = 20, recover_threshold: float = 1.0):
        """Initialize with configurable parameters for health regeneration."""
        self.max_health = max_health
        self.recover_threshold = recover_threshold

    def precondition(self, current_state: WorldState, action: str) -> bool:
        """
        Return True if the player's inventory health should regenerate.
        This law applies if the player is not at max health, has sufficient
        recover points, and is not sleeping.
        """
        player = current_state.player

        # Check if player's current inventory health is less than the defined
maximum
        has_space_for_health = player.inventory.health < self.max_health

        # Check if player has sufficient recover points to enable regeneration
        has_recover_points = player.recover >= self.recover_threshold

        # Check if the player is not currently sleeping
        not_sleeping = not player.sleeping

        # This is a passive regeneration effect, so the specific action taken (e.
g., "Move North")
        # is not a direct precondition, but the effect occurs during the state
transition.
        return has_space_for_health and has_recover_points and not_sleeping

    def effect(self, current_state: WorldState, action: str) -> None:
        """
        Apply the law by increasing the player's inventory health by 1.
        """
        # Increment the player's inventory health by 1.
        current_state.player.inventory.health = DiscreteDistribution(support=[
current_state.player.inventory.health + 1])
```

**Box B.6| Skeleton Idle**

```python
class SkeletonIdleLaw:
    def __init__(self):
        """Initialize with configurable parameters."""
        pass

    def precondition(self, current_state: WorldState, action: str) -> bool:
        """Return True if this law should apply to the given state and action."""
        # This law applies if there are any skeletons in the world that aren't
otherwise engaged.
        # Since no changes were observed, we assume this is their default passive
 behavior.
        return True # Applies universally as a default behavior for skeletons

    def effect(self, current_state: WorldState, action: str) -> None:
        """Apply the law by modifying the world state."""
        for skeleton in current_state.get_object_of_type_in_update_range(
SkeletonState):
            # Based on observation, skeletons remain unchanged.
```

```
16              # We predict their attributes will stay the same.
17              skeleton.health = DiscreteDistribution(support=[skeleton.health])
18              skeleton.position.x = DiscreteDistribution(support=[skeleton.position
    .x])
19              skeleton.position.y = DiscreteDistribution(support=[skeleton.position
    .y])
20              skeleton.reload = DiscreteDistribution(support=[skeleton.reload])
```

## C  THE CRAFTER-OO ENVIRONMENT

This appendix details Crafter-OO, our reimplementation of the Crafter environment that exposes a structured, object-oriented symbolic state and operates through a pure transition function. We developed Crafter-OO as a testbed for symbolic world modeling approaches in a complex, stochastic domain.

Table 3: The discrete action space of Crafter-OO. The action space is identical to the original Crafter benchmark (Hafner, 2022).

| Category | Actions |
|---|---|
| Movement | move_left, move_right, move_up, move_down |
| Interaction | do, sleep, noop |
| Placement | place_stone, place_table, place_furnace, place_plant |
| Crafting | make_wood_pickaxe, make_stone_pickaxe, make_iron_pickaxe make_wood_sword, make_stone_sword, make_iron_sword |

### C.1  MOTIVATION AND DESIGN PRINCIPLES

Symbolic world modeling benefits from environments where the complete state is accessible as a structured representation. Simple grid worlds provide this but lack complexity, while more complex environments typically require additional engineering to expose their internal state. More fundamentally, existing testbeds for symbolic world modeling have focused on environments that are either deterministic or have limited stochasticity and a narrow range of mechanics. Atari games, for instance, while complex in visual processing demands, have relatively predictable dynamics and a constrained set of interactions compared to open-world environments.

We developed Crafter-OO to address this gap. The environment features significant stochasticity in entity behaviors, diverse mechanics spanning resource collection to combat, and multi-step causal chains. Our design follows three principles:

1. **Explicit Object-Oriented State**: The entire game state is captured in a single, hierarchical data model that serves as input and output for world models.
2. **Functional Purity**: The environment's dynamics are exposed as a pure transition function, $T(\text{state}, \text{action}) \rightarrow \text{next\_state}$, with no hidden variables.
3. **Programmatic Modification**: The state representation can be precisely manipulated with code, enabling controlled experimental setups.

### C.2  THE WorldState DATA MODEL

The core of Crafter-OO is the WorldState data model, which captures the environment at a single timestep. This model is defined using Pydantic for structure and validation. Its components include:

- player: A PlayerState object containing position, inventory, health, and current action.
- objects: A list of non-player entities (CowState, ZombieState, PlantState, etc.) with type discrimination via a name field.
- materials: A 2D array representing the terrain map.

- **Global Properties**: World-level attributes including `daylight`, `size`, and serialized random state.

Listing 1 shows the structure of this model. This representation provides the interface between the environment and symbolic world models.

```python
from typing import TypeAlias, Literal

# --- Basic Data Structures ---

class Position:
    """Represents a 2D position (x, y) in the game world."""
    x: int
    y: int

class Inventory:
    """Represents the player's inventory counts for each item type."""
    health: int
    food: int
    drink: int
    energy: int
    sapling: int
    wood: int
    stone: int
    coal: int
    iron: int
    diamond: int
    wood_pickaxe: int
    stone_pickaxe: int
    iron_pickaxe: int
    wood_sword: int
    stone_sword: int
    iron_sword: int

class Achievements:
    """Represents the player's unlocked achievements."""
    collect_coal: int
    collect_diamond: int
    collect_drink: int
    collect_iron: int
    collect_sapling: int
    collect_stone: int
    collect_wood: int
    defeat_skeleton: int
    defeat_zombie: int
    eat_cow: int
    eat_plant: int
    make_iron_pickaxe: int
    make_iron_sword: int
    make_stone_pickaxe: int
    make_stone_sword: int
    make_wood_pickaxe: int
    make_wood_sword: int
    place_furnace: int
    place_plant: int
    place_stone: int
    place_table: int
    wake_up: int

# --- Game World Entities ---

class BaseObject:
    """The base class for all dynamic objects in the game world."""
    entity_id: int
    position: Position
```

```python
 61     health: int
 62     removed: bool
 63
 64 class Player(BaseObject):
 65     """The state of the player character."""
 66     name: Literal["player"] = "player"
 67     facing: Position
 68     action: str
 69     sleeping: bool
 70     inventory: Inventory
 71     achievements: Achievements
 72     thirst: float
 73     hunger: float
 74     fatigue: float
 75     recover: float
 76     last_health: int
 77
 78 class Cow(BaseObject):
 79     """The state of a cow."""
 80     name: Literal["cow"] = "cow"
 81
 82 class Zombie(BaseObject):
 83     """The state of a zombie."""
 84     name: Literal["zombie"] = "zombie"
 85     cooldown: int
 86
 87 class Skeleton(BaseObject):
 88     """The state of a skeleton."""
 89     name: Literal["skeleton"] = "skeleton"
 90     reload: int
 91
 92 class Arrow(BaseObject):
 93     """The state of an arrow projectile."""
 94     name: Literal["arrow"] = "arrow"
 95     facing: Position
 96
 97 class Plant(BaseObject):
 98     """The state of a plant, which can be eaten."""
 99     name: Literal["plant"] = "plant"
100     grown: int
101     ripe: bool
102
103 class Fence(BaseObject):
104     """The state of a fence object."""
105     name: Literal["fence"] = "fence"
106
107 # A union of all possible entity types in the world.
108 Entity: TypeAlias = Player | Cow | Zombie | Skeleton | Arrow | Plant | Fence
109
110
111 # --- World and Spatial Structures ---
112
113 MaterialT: TypeAlias = str
114
115 class Chunk:
116     """Represents a spatial region of the world for efficient updates."""
117     chunk_key: tuple[int, int, int, int]
118     object_ids: list[int]
119
120 class WorldState:
121     """Represents the complete, hierarchical state of the game world at a single
          timestep."""
122     # World dimensions and configuration
123     size: tuple[int, int]
124     chunk_size: tuple[int, int]
```

```
125     view: tuple[int, int]
126
127     # World status
128     daylight: float
129     step_count: int
130
131     # The grid of static materials (e.g., grass, stone, water)
132     materials: list[list[MaterialT | None]]
133
134     # A list of all dynamic entities currently in the world.
135     objects: list[Entity]
136
137     # A direct reference to the player object for easy access.
138     player: Player
139
140     # Spatial partitioning data.
141     chunks: list[Chunk]
142
143     # Internal simulation state
144     entity_id_counter_state: int
145     serialized_random_state: str
146     event_bus: list[str]
```

Listing 1: Simplified structure of the `WorldState` data structure.

## C.3 EXTRACTING STATE FROM CRAFTER'S GAME ENGINE

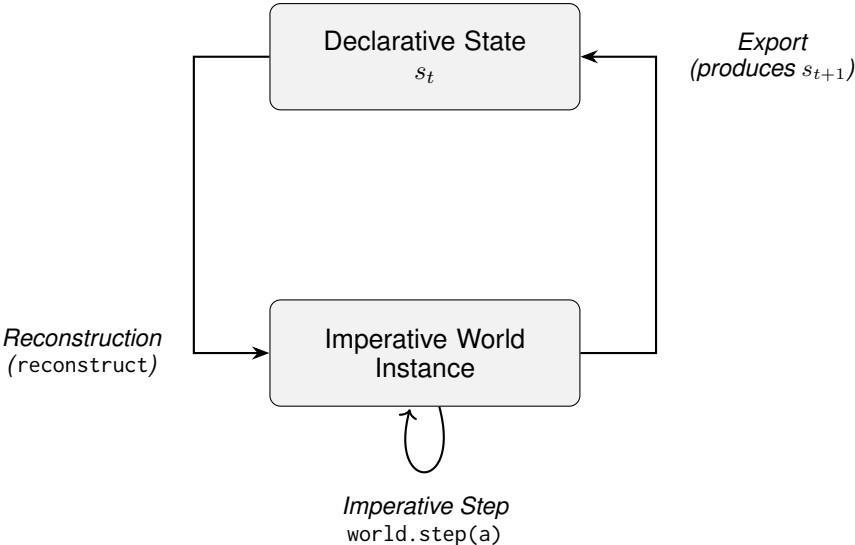

Figure 6: The functional cycle for state transition. A declarative state snapshot is reconstructed into a live, imperative world instance. The engine simulates a single step, and the resulting world is exported back into a new declarative state snapshot for the next timestep. This ensures we match Crafter's mechanics exactly.

The simulation state in the original engine is not a single data structure but is distributed across a graph of live Python objects, each with its own internal state and complex inter-dependencies, such as non-player characters holding direct references to the player object. Furthermore, the engine's behavior relies on implicit state, including the internal state of its pseudo-random number generator, which governs all stochastic events. Achieving a pure functional interface required developing a robust mechanism to first serialize this entire, complex state into a self-contained, declarative representation and then perfectly reconstruct the live object graph from that representation for each step of the simulation.

The state export process transforms the live simulation into a serializable snapshot. This procedure performs a deep traversal of the game engine's internal state, capturing all information required to reproduce the exact game moment. This includes the grid of world materials, the positions of all entities, and the type-specific attributes of each entity, such as a zombie's attack cooldown or a plant's growth progress. Crucially, the process also serializes the state of the engine's pseudo-random number generator, ensuring that the sequence of random numbers for subsequent stochastic events is preserved. To maintain the spatial partitioning data used for efficient queries, the set of entities within each world chunk is recorded by storing their unique identifiers. The final output is a complete, declarative data structure that represents the world at a single point in time, free from any live object references or other runtime-specific information.

State reconstruction reverses this process, rebuilding the live simulation from the declarative snapshot. This is more complex than simply loading data. It involves re-instantiating the entire graph of game objects and correctly re-establishing their inter-dependencies. A key complexity arises from object relationships; for instance, hostile entities require a direct reference to the live player object to guide their behavior. To resolve this, we employ a multi-pass reconstruction algorithm. First, entities with no external dependencies, such as the player, are instantiated. Then, dependent entities are instantiated in a second pass, receiving references to the already-created objects they require. Once all objects are created, the spatial partitioning system is rebuilt by mapping the stored entity identifiers back to the newly created live object instances. Finally, the deserialized state of the pseudo-random number generator is loaded, ensuring that the reconstructed world will produce the exact same stochastic outcomes as the original. The overall process is described in Box 1 and illustrated in Figure 1.

---

**Box C.1| Pseudocode for the Functional Transition Cycle**

```
function FunctionalTransition(declarative_state_t, action_t):
  // 1. Reconstruct the imperative world from the declarative state snapshot.
  world_instance <- ReconstructWorldFromState(declarative_state_t)

  // 2. Emulate a single step in the imperative engine.
  player <- FindPlayerObject(world_instance)
  ApplyActionToPlayer(player, action_t)
  for object in world_instance.get_all_objects():
    object.update()

  // 3. Export the new world state into a declarative representation.
  declarative_state_t+1 <- ExportStateFromWorld(world_instance)

  return declarative_state_t+1

function ExportStateFromWorld(world_instance):
  snapshot <- new DeclarativeState
  snapshot.materials <- CopyGrid(world_instance.material_grid)
  snapshot.rng_state <- Serialize(world_instance.random_generator)
  for object in world_instance.get_all_objects():
    AddObjectState(snapshot, object.type, object.attributes, object.id)
  return snapshot

function ReconstructWorldFromState(snapshot):
  world_instance <- new ImperativeWorld
  world_instance.material_grid <- CopyGrid(snapshot.materials)
  world_instance.random_generator <- Deserialize(snapshot.rng_state)

  // Multi-pass object instantiation to handle dependencies.
  player_state <- FindPlayerStateInSnapshot(snapshot)
  player_object <- InstantiateObject(
    player_state.type, player_state.attributes
```

```
    )
    AddObjectToWorld(world_instance, player_object)

    for object_state in snapshot.get_all_object_states():
      if not is_player(object_state):
        // Pass player reference to dependent objects (e.g., Zombie).
        dependencies <- {player: player_object}
        new_object <- InstantiateObject(
          object_state.type, object_state.attributes, dependencies
        )
        AddObjectToWorld(world_instance, new_object)

    RebuildSpatialIndex(world_instance)
    return world_instance
```

## C.4 THE FUNCTIONAL ENVIRONMENT INTERFACE

We provide a `transition` function that implements a stateless API for environment steps:

1. Input: `WorldState` object $s_t$
2. Reconstruct live game engine instance
3. Execute single update tick with given action
4. Export resulting state as $s_{t+1}$
5. Return new `WorldState` object

This ensures every transition is a pure function of the explicit state, making the environment suitable for symbolic reasoning and program synthesis.

## C.5 UTILITIES FOR PROGRAMMATIC STATE INTERACTION

A key contribution of Crafter-OO is a rich set of utilities that enable programmatic interaction with the world state. These functions are essential for two purposes: first, they allow for the precise, reproducible setup of the evaluation scenarios discussed in Section E; second, they provide a high-level API that simplifies the authoring of programmatic world model laws. To provide a clear overview of this toolkit, Table 4 catalogues the key functions, which are grouped into three main categories: World Setup, Player State, and High-Level State Queries & Modifications.

Table 4: A catalogue of key utilities for programmatic state manipulation in Crafter-OO. These functions provide the building blocks for creating controlled experimental scenarios and for writing concise, high-level world model laws.

| Category | Function Signature (Simplified) | Description |
|---|---|---|
| **World Setup Utilities** | set_tile_material(pos, material) | Modifies the terrain at a specific coordinate (e.g., changes grass to stone). |
| | add_object_to_world(cls, pos, ...) | Adds an entity instance (e.g., a Cow or Zombie) to the world. |
| | remove_object_from_world(obj) | Removes a specific entity instance from the world. |
| | set_daylight(level) | Sets the global daylight level, affecting visibility and mob spawning. |
| **Player State Utilities** | set_player_position(pos) | Sets the player's exact (x, y) coordinates. |
| | set_player_facing(direction) | Sets the player's facing direction (e.g., up, down, left, right). |
| | set_player_inventory_item(item, qty) | Sets the quantity of a specific item in the player's inventory. |
| | set_player_internal_stat(stat, val) | Adjusts internal player stats like health, hunger, or energy. |
| **High-Level State Queries & Modifications** | get_target_tile() | Returns the material and any object at the tile the player is facing. |
| | get_object_of_type_in_update_range(cls) | Returns all entities of a specific type near the player. |
| | move_object(obj, dir, walkable) | Moves an entity one step if the target tile is valid and unoccupied. |
| | set_facing_material(material) | Changes the material of the tile the player is facing. |

These utilities are composed to construct the specific initial conditions for our evaluation scenarios. Listing 2 demonstrates how they work in concert to create a test case for a resource collection mechanic. World setup utilities are first used to clear an area and place a specific resource (coal). Then, player state utilities are used to position the player correctly and provide the necessary tool (wood_pickaxe) in their inventory. This level of programmatic control, enabled by the functions detailed in Table 4, is what makes our targeted evaluation methodology possible.

```
1  def get_initial_state_for_coal_collection():
2      # Create a base world and get references to the world and player objects
3      world = reconstruct_world_from_state(initial_state())
4      player = find_player(world)
5
6      # --- World Setup Utilities ---
7      # Clear a 3x3 area around the player to be grass
8      for x in range(4, 7):
9          for y in range(4, 7):
10             world_utils.set_tile_material(world, (x, y), "grass")
11
12     # Place the target resource in a specific location
13     world_utils.set_tile_material(world, (6, 5), "coal")
14
15     # --- Player State Utilities ---
16     # Set the player's starting position
17     player_utils.set_player_position(player, (5, 5))
18
19     # Make the player face the target resource
20     player_utils.set_player_facing(player, (1, 0))
21
22     # Add the required tool to the player's inventory
23     player_utils.set_player_inventory_item(player, "wood_pickaxe", 1)
24
25     # Convert the configured world back to a serializable WorldState
26     return export_world_state(world, view=(9, 9))
```

Listing 2: Example of programmatic state manipulation to create an initial state for a scenario. World setup utilities create the environment, while player state utilities configure the agent.

## D  MUTATORS

Mutators are a core component of our evaluation framework, designed to test a world model's ability to distinguish between plausible and implausible future states, as described in Sec. 4. A mutator is a deterministic function that takes a state-action pair $(s_t, a_t)$ and produces an alternative, incorrect next state $\tilde{s}_{t+1}$. These generated states, called distractors, represent violations of the environment's true dynamics. For example, a distractor might show the agent crafting an item without the necessary resources or moving through a solid obstacle.

By creating a candidate set containing the true next state $s_{t+1}$ and several such distractors $\{\tilde{s}_{t+1}\}$, we construct a discriminative task for the world model. A model with a robust understanding of the environment's laws should assign a significantly higher probability to the true outcome than to any of the distractors. This allows us to quantitatively measure the model's predictive judgment using the state ranking metrics from Sec. 4.

All mutators adhere to a common interface, shown in Listing 3. Each mutator implements a 'precondition' method that checks if the mutation is applicable to a given state and action. If the precondition is met, the 'effect' method is called to generate the mutated state. This design allows for the creation of targeted mutators that only apply under specific circumstances, leading to more subtle and challenging distractors.

```
1  class Mutator:
2      """A protocol for functions that generate distractor states."""
3
4      def precondition(self, state: WorldState, action: Action) -> bool:
5          """
6          Returns True if the mutator can be applied to the given
7          state-action pair, False otherwise.
8          """
9          ...
10
11     def __call__(self, state: WorldState, action: Action) -> WorldState:
12         """
```

```
13          Applies a mutation to a copy of the state and returns the
14          modified state, representing an illegal transition outcome.
15          """
16          ...
```

Listing 3: The general interface for a mutator. Each mutator is a callable object with a method to check for applicability.

We have implemented a suite of mutators for the Crafter-OO environment, categorized by the type of game mechanic they target. Tab. 5 provides a comprehensive list of these mutators and the specific rule violations they introduce.

Table 5: Catalogue of mutators implemented for the Crafter-OO environment.

| Category | Mutator Name | Description of Rule Violation |
|---|---|---|
| Physics | IllegalMovementMutator | Causes the player to move when a non-movement action is taken. |
| | EntityPositionMutator | Teleports non-player entities to random distant locations. |
| Combat | PlayerHealthMutator | Arbitrarily adds or subtracts a small amount of health from the player. |
| | EntityHealthMutator | Sets the health of non-player entities to a random, incorrect value. |
| Crafting | CraftIllegalItemMutator | Produces a different item than the one specified by the crafting action. |
| Collection | CollectIllegalMaterialMutator | Adds an incorrect resource to the player's inventory when collecting. |
| Placement | PlaceIllegalItemMutator | Places a different object or tile than the one specified by the action. |
| Player State | InventoryMutator | Randomizes all quantities in the player's inventory. |

Below we provide detailed descriptions and simplified implementations for three representative mutators from different categories.

ILLEGAL MOVEMENT MUTATOR

This mutator tests the model's understanding of which actions cause player movement. It activates when the agent takes an action that should not result in a change of position, such as noop or do. The effect is to move the player one step in a random direction, creating a state that would be valid for a movement action but is invalid for the action actually taken. Listing 4 shows its logic.

```python
1  NON_MOVEMENT_ACTIONS = {"noop", "do", "sleep", "make_wood_pickaxe", ...}
2  DIRECTIONS = [(0, 1), (1, 0), (0, -1), (-1, 0)]
3
4  class IllegalMovementMutator:
5      def precondition(self, state: WorldState, action: Action) -> bool:
6          # This mutator applies only to actions that should not cause movement.
7          return action in NON_MOVEMENT_ACTIONS
8
9      def __call__(self, state: WorldState, action: Action) -> WorldState:
10         mutated_state = state.model_copy(deep=True)
11
12         # Choose a random direction and update the player's position.
13         random_direction = random.choice(DIRECTIONS)
14         mutated_state.player.position.x += random_direction[0]
15         mutated_state.player.position.y += random_direction[1]
16
17         return mutated_state
```

Listing 4: Simplified logic for the IllegalMovementMutator.

CRAFT ILLEGAL ITEM MUTATOR

This mutator targets the logic of crafting recipes. It checks if the agent is attempting to craft an item. If so, it alters the outcome by giving the player a different, randomly selected craftable item. This tests whether the world model has correctly associated specific crafting actions with their unique outcomes. For example, if the action is make_wood_pickaxe, this mutator might instead add a stone_sword to the player's inventory. Listing 5 illustrates this process.

```
1  CRAFTING_ACTIONS = {"make_wood_pickaxe", "make_stone_sword", ...}
2
3  class CraftIllegalItemMutator:
4      def precondition(self, state: WorldState, action: Action) -> bool:
5          # This mutator applies only to crafting actions.
6          return action in CRAFTING_ACTIONS
7
8      def __call__(self, state: WorldState, action: Action) -> WorldState:
9          mutated_state = state.model_copy(deep=True)
10
11         # Select a different crafting action to determine the illegal outcome.
12         other_crafting_actions = CRAFTING_ACTIONS - {action}
13         illegal_action = random.choice(list(other_crafting_actions))
14
15         # Add the item corresponding to the illegal action to the inventory.
16         if illegal_action == "make_stone_sword":
17             mutated_state.player.inventory.stone_sword += 1
18         # ... logic for other craftable items
19
20         return mutated_state
```

Listing 5: Simplified logic for the `CraftIllegalItemMutator`.

ENTITY HEALTH MUTATOR

This mutator introduces arbitrary changes to the health of non-player characters (NPCs), violating the rules of combat, regeneration, and damage. It is an "always on" mutator, meaning its precondition is always true, as health can be a dynamic property in any state. Its effect is to iterate through all non-player entities and set their health to a random value that is not close to their current health. This prevents generating trivial changes that might occur naturally (e.g., from regeneration) and creates a more distinctively incorrect state. Listing 6 shows the implementation.

```
1  class EntityHealthMutator:
2      def precondition(self, state: WorldState, action: Action) -> bool:
3          # This mutator is always applicable.
4          return True
5
6      def __call__(self, state: WorldState, action: Action) -> WorldState:
7          mutated_state = state.model_copy(deep=True)
8
9          for entity in mutated_state.objects:
10             # Skip the player entity.
11             if entity.entity_id == mutated_state.player.entity_id:
12                 continue
13
14             # Generate a new health value that is not the same as the current
15             # health, nor immediately adjacent to it.
16             possible_health_values = set(range(11)) # Health is 0-10
17             excluded_values = {entity.health, entity.health - 1, entity.health + 1}
18             valid_new_values = list(possible_health_values - excluded_values)
19
20             if valid_new_values:
21                 entity.health = random.choice(valid_new_values)
22
23         return mutated_state
```

Listing 6: Simplified logic for the `EntityHealthMutator`.

# E  SCENARIOS

An evaluation framework that relies on data from unguided exploration may not sufficiently cover all of an environment's mechanics, especially those that are rare or require specific preconditions.

To ensure a comprehensive and targeted assessment of a world model's understanding, we generate evaluation data from a suite of **scenarios**. Each scenario is a short, programmatic interaction sequence designed to isolate and test a single game mechanic under controlled conditions. This approach produces a dataset of transitions that robustly covers the environment's dynamics, from basic resource collection to complex combat encounters. The transitions generated by these scenarios form the basis for the evaluation metrics described in Sec. 4.

### E.1 SCENARIO STRUCTURE AND EXECUTION

A scenario is defined by a common programmatic interface, as outlined in listing 7. It specifies an initial state, a scripted policy to guide the agent's actions, and a termination condition based on either achieving a specific goal or reaching a maximum number of steps. The execution of a scenario, shown in listing 8, produces a sequence of (state, action, next_state) transitions that serve as ground truth test cases for the world model.

```
class Scenario:
    @property
    def name(self) -> str: ...

    def get_initial_state(self) ->
    WorldState: ...

    def policy(self, state: WorldState) ->
     Action: ...

    def goal_test(self, transitions: list)
     -> bool: ...

    @property
    def max_steps(self) -> int: ...
```

Listing 7: Structure of an evaluation scenario.

```
def run_scenario(scenario):
    transitions = []
    state = scenario.get_initial_state()
    for _ in range(scenario.max_steps):
        action = scenario.policy(state)
        next_state = env.transition(state,
     action)
        transitions.append((state, action,
     next_state))
        state = next_state
        if scenario.goal_test(transitions)
     :
            break
    return transitions
```

Listing 8: Execution loop for generating transitions.

### E.2 IMPLEMENTED SCENARIOS

We developed over 40 scenarios for Crafter-OO, covering every core game mechanic present in the original Crafter environment. These scenarios are categorized by the type of mechanic they test, as detailed in Tab. 6. For many mechanics, we include both a "successful" and an "unsuccessful" variant. The successful version sets up the preconditions for an action to succeed (e.g., having enough resources to craft an item), while the unsuccessful version deliberately violates a precondition. This allows us to test whether a world model understands not only what should happen, but also what should *not* happen.

## F EVALUATION IMPLEMENTATION DETAILS

This section provides a procedural specification of our evaluation framework. We begin by defining a general-purpose interface that any world model must satisfy to be evaluated. We then detail the computational steps that transform the raw outputs of a model satisfying this interface into the final State Fidelity and State Ranking metrics presented in Sec. 4. The process relies on the evaluation trajectories generated from Scenarios (Sec. E) and the distractor states generated by Mutators (Sec. D).

Our evaluation framework is designed to be model-agnostic. Any world model can be benchmarked, provided it adheres to the simple, two-method interface shown in listing 9. This interface cleanly separates the two core capabilities required for our metrics: the ability to generate a likely future state (for fidelity) and the ability to score a given future state (for ranking).

```
class EvaluatableWorldModel(Protocol):
    """A protocol for world models that can be evaluated by our framework."""

```

Table 6: Complete list of evaluation scenarios used to test world models in Crafter-OO.

| Category | Scenario Name | Description |
|---|---|---|
| **Movement** | random_movement | Tests basic player movement in the cardinal directions. |
| **Collection** | collect_wood | Player faces a tree and collects wood. |
| | collect_drink | Player faces water and collects it. |
| | collect_stone | Player collects stone with the required pickaxe. |
| | unsuccessful_collect_stone | Player attempts to collect stone without the required pickaxe. |
| | collect_coal | Player collects coal with the required pickaxe. |
| | unsuccessful_collect_coal | Player attempts to collect coal without the required pickaxe. |
| | collect_iron | Player collects iron with the required pickaxe. |
| | unsuccessful_collect_iron | Player attempts to collect iron without the required pickaxe. |
| | collect_diamond | Player collects diamond with the required pickaxe. |
| | unsuccessful_collect_diamond | Player attempts to collect diamond without the required pickaxe. |
| | eat_plant | Player eats a ripe plant to gain food. |
| | unsuccessful_eat_plant | Player attempts to eat an unripe plant. |
| **Crafting** | craft_wooden_pickaxe | Player crafts a wooden pickaxe with sufficient wood. |
| | unsuccessful_craft_wooden_pickaxe | Player attempts to craft without sufficient wood. |
| | craft_wooden_sword | Player crafts a wooden sword with sufficient wood. |
| | unsuccessful_craft_wooden_sword | Player attempts to craft without sufficient wood. |
| | craft_stone_pickaxe | Player crafts a stone pickaxe with required resources. |
| | unsuccessful_craft_stone_pickaxe | Player attempts to craft without required resources. |
| | craft_stone_sword | Player crafts a stone sword with required resources. |
| | unsuccessful_craft_stone_sword | Player attempts to craft without required resources. |
| | craft_iron_pickaxe | Player crafts an iron pickaxe with required resources. |
| | unsuccessful_craft_iron_pickaxe | Player attempts to craft without required resources. |
| | craft_iron_sword | Player crafts an iron sword with required resources. |
| | unsuccessful_craft_iron_sword | Player attempts to craft without required resources. |
| **Placement** | place_table | Player places a crafting table with sufficient wood. |
| | unsuccessful_place_table | Player attempts to place a table without sufficient wood. |
| | place_stone | Player places stone with sufficient inventory. |
| | unsuccessful_place_stone | Player attempts to place stone without sufficient inventory. |
| | place_furnace | Player places a furnace with sufficient stone. |
| | unsuccessful_place_furnace | Player attempts to place a furnace without sufficient stone. |
| | place_plant | Player places a sapling on a grass tile. |
| | unsuccessful_place_plant | Player attempts to place a sapling without one in inventory. |
| **Combat** | zombie_defeat | Player, equipped with a sword, defeats a zombie. |
| | defeat_skeleton | Player defeats a skeleton. |
| | eat_cow | Player defeats a cow to obtain food. |
| | player_death | Player with low health is defeated by a zombie. |
| **NPC Behavior** | cow_movement | Tests the stochastic movement of a cow over several steps. |
| | wake_up | Player goes to sleep and wakes up after their energy is restored. |

```python
def sample_next_state(self, current_state: WorldState, action: Action) -> WorldState:
    """
    Generative function: Samples a single predicted next state s_hat_{t+1}
    from the model's posterior distribution P(s_{t+1} | s_t, a_t).
    """
    ...

def evaluate_log_probability(
    self, state: WorldState, action: Action, next_state: WorldState
) -> float:
    """
    Discriminative function: Computes the log-probability of a specific
    next_state given the current state and action.
    """
    ...
```

Listing 9: The interface any world model must implement to be compatible with our evaluation framework.

### F.1 STATE COMPARISON VIA CANONICAL REPRESENTATION

All metrics that involve comparing two world states, such as edit distance or checking for equality, require a deterministic and canonical representation of the state. A direct object-to-object comparison can be unreliable due to factors like in-memory object identifiers or the ordering of elements in lists. To address this, we serialize each `WorldState` object to a canonical JSON format before any comparison is performed. This process, outlined in listing 10, ensures that two states are considered identical if and only if they represent the same game-world configuration.

```python
def to_canonical_json(state: WorldState) -> dict:
    """
    Serializes a WorldState object to a deterministic JSON representation.
    """
    # 1. Exclude non-semantic or non-deterministic fields from serialization.
    excluded_fields = {"event_bus", "serialized_random_state"}
    serialized_state = state.model_dump(exclude=excluded_fields, mode="json")

    # 2. Sort lists of objects by a stable, unique key to ensure order invariance.
    # The player object is handled separately and removed from the main list.
    serialized_state["objects"] = [
        obj for obj in serialized_state["objects"] if obj["name"] != "player"
    ]
    serialized_state["objects"].sort(key=lambda obj: obj["entity_id"])

    # Chunks are also sorted to ensure map representation is stable.
    if "chunks" in serialized_state:
        serialized_state["chunks"].sort(key=lambda chunk: chunk["chunk_key"])

    return serialized_state
```

Listing 10: Canonical serialization of a WorldState object.

### F.2 STATE FIDELITY METRIC CALCULATION

The state fidelity metrics measure the difference between a world model's predicted next state and the ground truth. We use JSON Patch (Bryan & Nottingham, 2013), a standard for describing changes in a JSON document, to provide a precise, interpretable measure of this difference. The calculation for a single transition $(s_t, a_t, s_{t+1})$ proceeds as described in listing 11.

```python
def calculate_state_fidelity(world_model, s_t, a_t, s_t_plus_1):
    """
    Computes Raw and Normalized Edit Distance for a world model's prediction.
    """
    # 1. Generate a predicted next state from the world model.
    s_hat_t_plus_1 = world_model.sample_next_state(s_t, a_t)

    # 2. Convert both true and predicted next states to canonical JSON.
    json_true = to_canonical_json(s_t_plus_1)
    json_predicted = to_canonical_json(s_hat_t_plus_1)

    # 3. Compute the JSON Patch from the predicted state to the true state.
    patch = jsonpatch.make_patch(json_predicted, json_true)

    # 4. Raw Edit Distance is the number of operations in the patch.
    raw_edit_distance = len(list(patch))

    # 5. Normalized Edit Distance is the raw distance divided by the total number
    # of elements in the true state, providing a scale-invariant measure.
    total_elements = count_elements(json_true)
    normalized_edit_distance = raw_edit_distance / total_elements if total_elements > 0
     else 0

    return raw_edit_distance, normalized_edit_distance
```

Listing 11: Calculation of State Fidelity metrics for a single transition.

**Example.** Consider a transition where the player, at position $(x = 5, y = 5)$ with $health = 9$, takes the action move_right. The true next state, $s_{t+1}$, has the player at $(x = 6, y = 5)$ with $health = 9$. Suppose a world model predicts a state, $\hat{s}_{t+1}$, where the player correctly moves to $(x = 6, y = 5)$ but their health incorrectly drops to $8$.

The simplified canonical JSON representations for the player object in each state would be:

```
1 {
2    "player": {
3      "position": {"x": 6, "y": 5},
4      "health": 9
5    }
6 }
```

Listing 12: Canonical JSON for the true next state.

```
1 {
2    "player": {
3      "position": {"x": 6, "y": 5},
4      "health": 8
5    }
6 }
```

Listing 13: Canonical JSON for the predicted next state.

The JSON Patch required to transform the predicted JSON into the true JSON is a single replace operation: [{''op'': ''replace'', ''path'': ''/player/health'', ''value'': 9}]. The Raw Edit Distance is the number of operations in this patch, which is 1. The Normalized Edit Distance would be this value divided by the total number of elements in the true state's full JSON representation.

### F.3 STATE RANKING METRIC CALCULATION

State ranking metrics evaluate a model's ability to distinguish the true outcome of an action from a set of plausible but incorrect alternatives. This process involves generating a set of candidate states and using the world model to score them, as detailed in listing 14.

```python
def calculate_state_ranking(world_model, s_t, a_t, s_t_plus_1, mutators, num_distractors
    ):
    """
    Computes Rank@1 and Mean Reciprocal Rank for a world model.
    """
    # 1. Generate a set of distractor states using the mutator bank.
    distractors = []
    applicable_mutators = [m for m in mutators if m.precondition(s_t, a_t)]
    random.shuffle(applicable_mutators) # Ensure variety in distractors
    for mutator in applicable_mutators:
        if len(distractors) >= num_distractors:
            break
        distractors.append(mutator(s_t, a_t))

    # 2. Form the candidate set, including the ground truth and distractors.
    candidate_set = [s_t_plus_1] + distractors
    random.shuffle(candidate_set) # Avoid biasing models that may be sensitive to order

    # 3. Score each candidate state using the world model's log-probability function.
    scores = []
    for s_candidate in candidate_set:
        log_prob = world_model.evaluate_log_probability(s_t, a_t, s_candidate)
        scores.append(log_prob)

    # 4. Determine the rank of the true next state.
    # Ranks are 1-indexed, with rank 1 being the highest score.
    ranked_indices = sorted(range(len(scores)), key=lambda i: scores[i], reverse=True)
    true_state_index = candidate_set.index(s_t_plus_1)
    rank_of_true_state = ranked_indices.index(true_state_index) + 1

    # 5. Calculate metrics from the rank.
    rank_at_1 = 1.0 if rank_of_true_state == 1 else 0.0
    reciprocal_rank = 1.0 / rank_of_true_state
```

```
34    return rank_at_1, reciprocal_rank
```

Listing 14: Calculation of State Ranking metrics for a single transition.

**Example.** Continuing the previous example, the true state $s_{t+1}$ is the player moving right. A mutator might generate a distractor state $s_{\text{distractor}}$ where the player illegally teleports to $(x = 20, y = 20)$. The candidate set becomes $\{s_{t+1}, s_{\text{distractor}}\}$. A good world model should assign a much higher probability to the true outcome. For instance, it might yield log-probabilities of $\log p(s_{t+1}|\dots) = -0.7$ and $\log p(s_{\text{distractor}}|\dots) = -15.4$. Since $-0.7 > -15.4$, the true state is ranked first. This yields a Rank@1 of 1.0 and a Mean Reciprocal Rank of $1/1 = 1.0$ for this transition.

### F.4 AGGREGATION ACROSS SCENARIOS

The final metrics reported in Tab. 1 are aggregated from the per-transition results. To ensure that each distinct game mechanic contributes equally to the final score, we employ a two-level aggregation strategy. First, we compute the mean metric values across all transitions within a single scenario. Second, we compute the final reported metric by taking the mean of these per-scenario means. This prevents scenarios with more transitions (e.g., a long movement sequence) from dominating the overall results compared to scenarios with fewer, more critical transitions (e.g., a single crafting action). listing 15 formalizes this entire pipeline.

```
1  def evaluate_world_model(world_model, scenarios, mutators, config):
2      """
3      Runs the full evaluation pipeline and returns aggregated metrics.
4      """
5      per_scenario_metrics = {}
6
7      # 1. Evaluate each scenario independently.
8      for scenario in scenarios:
9          transitions = run_scenario(scenario) # See Sec. C.1 for run_scenario
10
11         scenario_results = []
12         for (s_t, a_t, s_t_plus_1) in transitions:
13             # Calculate metrics for each transition in the scenario.
14             r_at_1, mrr = calculate_state_ranking(
15                 world_model, s_t, a_t, s_t_plus_1, mutators, config.num_distractors
16             )
17             raw_ed, norm_ed = calculate_state_fidelity(
18                 world_model, s_t, a_t, s_t_plus_1
19             )
20             scenario_results.append({
21                 "R@1": r_at_1, "MRR": mrr,
22                 "RawEditDist": raw_ed, "NormEditDist": norm_ed
23             })
24
25         # 2. First level of aggregation: average metrics within the scenario.
26         if not scenario_results: continue
27         per_scenario_metrics[scenario.name] = {
28             key: sum(res[key] for res in scenario_results) / len(scenario_results)
29             for key in scenario_results[0]
30         }
31
32     # 3. Second level of aggregation: average the per-scenario means.
33     final_metrics = {
34         key: sum(metrics[key] for metrics in per_scenario_metrics.values()) / len(
    per_scenario_metrics)
35         for key in list(per_scenario_metrics.values())[0]
36     }
37
38     return final_metrics
```

Listing 15: Overall evaluation pipeline and metric aggregation.

## G  SYNTHESIS AND EXPLORATION IMPLEMENTATION DETAILS

The process of generating candidate world laws is divided into two main stages: unguided exploration to collect a dataset of interactions, and law synthesis to propose programmatic laws from that dataset.

### G.1  EXPLORATION POLICY

To gather the interaction dataset $\mathcal{D} = \{(s_t, a_t, s_{t+1})\}_{t=1}^{N}$, we employ an autonomous exploration policy driven by a large language model. This policy operates without access to environment-specific rewards or human-provided goals. Instead, it is given a high-level instruction to explore the environment and discover as many of its underlying mechanics as possible, treating the task as a reverse-engineering problem. The full prompt provided to the exploration policy is detailed in box G.1.

---

**Box  G.1| Exploration Policy Prompt**

```
 1  You are an explorer in an unknown digital world. Your mission is to experience as
        many of the world's hidden mechanics as possible. Your recorded experiences
        will be analyzed later to create a complete map of the world's physical
        laws.
 2
 3  The laws of any world can be thought of as IF-THEN hypotheses: `IF (a specific
        situation occurs) AND (you take an ACTION), THEN (a certain outcome happens)
        .`
 4
 5  To succeed, you must trigger as many different `IF-THEN` scenarios as you can.
 6
 7  **What to Expect in the World:**
 8  This world is complex and may be dangerous.
 9  - **Hostile Entities:** You may encounter creatures that are hostile and will
        attack you.
10  - **Resource Collection:** The world contains raw materials that can be gathered,
         though there may be preconditions for collection.
11  - **Item Production:** You have the ability to craft useful items from raw
        materials, though there may be preconditions for production.
12  - **Combat:** You can engage in combat with the entities you encounter.
13
14  Your primary goal is to discover the rules governing these activities.
15  You will need to explore the game world by moving around and interacting with the
         entities and materials in the world.
16  If an action has no effect, you may not have fulfilled the preconditions for the
        action to have an effect.
17  Try out a variety of actions from each category: movement, interaction, placement
        , production.
18  If an action seems to have no effect, you may not have fulfilled the
        preconditions for the action to have an effect.
19  Try to acquire additional resources or change something about the world and try
        again.
20  Before taking actions, set goals for yourself in an IF-THEN format, and let the
        results invalidate those actions.
21  If an entity is hostile, you can attempt to defend yourself from it.
22  If an entity seems passive or beneficial, you can attempt to interact with it.
23  You will likely need to progress through the "tech tree" of the game in a
        specific order.
24  This will require interleaving resource collection with placement of crafting
        stations and production of better tools.
25  In the meantime, you will need to survive hostile enemies and find ways to heal
        from damage you've taken.
26  Some resources likely cannot be acquired without first producing a tool to
        acquire them.
```

---

```
27 Tools may require a mix of materials and crafting stations to produce.
28
29 The following are the only valid actions you can take:
30
31 {action_strings}.
32
33 You will now receive observations from the world. Begin your exploration.
```

This LLM-based policy is crucial for gathering sufficiently diverse data in a hostile environment like Crafter-OO. A purely random policy survives for an average of 100 steps before the agent perishes. In contrast, our LLM-based policy navigates the environment for an average of 400 steps. Despite this improvement, exploration remains a significant bottleneck. The policy often struggles to progress through the environment's technology tree, frequently failing to discover the necessary preconditions for crafting advanced items. It also exhibits a tendency to forget previously learned information, which prevents it from effectively building upon past successes within a single trajectory.

### G.2 LAW SYNTHESIS FROM TRAJECTORIES

The law synthesis pipeline processes the trajectory data from the exploration phase to generate a set of candidate laws $\{L_i\}$. The core idea is to identify state transitions where meaningful changes occur, and then prompt a large language model to propose atomic, programmatic laws that explain those specific changes. This process is outlined in Algorithm 17.

**Change Detection for Tractable Synthesis.** In an environment with a complex, structured state like Crafter-OO, changes between timesteps are often sparse and localized to specific sub-components. To make law synthesis tractable, we first isolate these localized changes to provide a focused context for the synthesizer. This is achieved through a set of detectors that monitor different **aspects** of the world state. An aspect is a semantically-cohesive subset of the state, typically corresponding to a top-level attribute (e.g., 'player.inventory') or a collection of entities of the same type (e.g., all 'ZombieState' objects). For each transition $(s_t, a_t, s_{t+1})$, we check for changes across all aspects. If a detector identifies a change, a synthesis task is created for that specific transition and aspect.

```python
class ChangeDetector:
    def aspect_name(self) -> str: ...
    def has_changes(self, s_t: WorldState, s_t_plus_1: WorldState) -> bool: ...

class PlayerInventoryChangeDetector(ChangeDetector):
    def aspect_name(self): return "player_inventory"
    def has_changes(self, s_t, s_t_plus_1):
        return s_t.player.inventory != s_t_plus_1.player.inventory

class ZombieStateChangeDetector(ChangeDetector):
    def aspect_name(self): return "zombies"
    def has_changes(self, s_t, s_t_plus_1):
        # Logic to compare zombie states between s_t and s_t_plus_1
        ...

# A list of all detectors is used to check each transition
ALL_DETECTORS = [
    PlayerInventoryChangeDetector(),
    ZombieStateChangeDetector(),
    ... # Other detectors for map tiles, cows, etc.
]
```

Listing 16: Simplified change detection logic. Each detector checks for changes in a specific part of the world state between $s_t$ and $s_{t+1}$.

This decomposition is not a form of environment-specific guidance but rather a generic mechanism derived directly from the structure of the state representation itself. The Crafter-OO environment

exposes an object-oriented state, defined by a schema of classes and attributes. Our change detectors mirror this schema, creating one detector for each top-level attribute and for each object type. This approach provides a structural inductive bias – that the environment's causal mechanisms are likely aligned with its object-oriented structure – without embedding knowledge of the environment's actual dynamics. The process could be fully automated for any environment that exposes a typed, structured state; the detectors can be generated programmatically by reflecting on the state schema. This is analogous to how a computer vision model might process distinct objects in a scene separately; we partition the state space based on its given structure, but the rules governing the interactions between these partitions must still be learned from scratch.

**Prompt Generation.** For each transition-aspect pair that triggers a synthesis task, we generate a detailed prompt for the LLM. The goal is to provide all necessary context for the model to infer the underlying game mechanic. The prompt contains several key components:

1. The initial state $s_t$ and resulting state $s_{t+1}$, serialized to a structured format (JSON).
2. The action $a_t$ that caused the transition.
3. A textual 'diff' that highlights the exact changes between $s_t$ and $s_{t+1}$.
4. A human-readable 2D ASCII rendering of the local environment around the player for both states, providing spatial context.
5. The name of the aspect (e.g., "player_inventory") that changed, which instructs the LLM to focus its analysis.

This structured presentation of the transition allows the LLM to ground its reasoning in the specific, observed changes. The full prompt template is provided in box G.2.

---

**Box G.2 | Synthesis Prompt**

```
1  ## Role
2  You are a **World Law Synthesizer** - an expert at analyzing game state
       transitions and extracting the underlying rules that govern virtual worlds.
       Your job is to observe how actions transform game states and codify these
       transformations into precise, executable laws that can model game mechanics,
        as well as try to model aspects of the underlying transition dynamics as
       functions.
3
4  ## Task Description
5  Given a world state, an action taken, an aspect of the state we are interested in
        modeling, and the resulting next world state (plus a diff highlighting the
       changes), you must:
6  - Identify how the aspect of the state we are interested in modeling changed
       between the observations
7  - Determine the underlying rules or laws that caused these changes
8  - Implement these laws as executable Python code using the provided WorldState
       interface and DiscreteDistribution for predictions
9
10 **IMPORTANT: You should write MULTIPLE laws when you observe multiple distinct
       changes.** Each law you write should be modular, minimalistic, focused on a
       single game mechanic, and capable of being combined with other laws to model
        complex game behavior.
11
12 In particular, you should strive to write laws that are responsible for as little
        of the state as possible. In any given transition, you may see many changes
       . Each of these changes could be caused by a different law. Think about what
        changes could be grouped together into a single law, and write separate
       laws for different types of changes.
13
14 - Break up the laws to each account for a single precondition and effect. For
       example, if an entity moves, write a law for the movement of entities of
       that type. If a player takes a particular action, write a law for that
       action specifically.
```

---

```
15  - Certain attributes cannot have a `DiscreteDistribution` applied to them. For
         example, the `materials` field should just be modified directly, not wrapped
          in a `DiscreteDistribution`. Alternatively, use `set_material` or `
         set_facing_material` to modify the materials field. Either way, they cannot
         be wrapped in a `DiscreteDistribution`.
16  - Use the `DiscreteDistribution` class to indicate probabilistic predictions, for
          example when trying to write a general law governing all entities of a type
          when you cannot reconcile all changes visible to that entity type into a
         deterministic law.
17  - You DO NOT need to use imports. Everything you need can be coded without the
         use of imports, and all classes defined below are already imported.
18
19  ## Aspect of the State
20  You will be given an aspect of the state we are interested in modeling. The laws
         you write should be focused on modeling changes to this aspect of the state.
21  However, you can use _all_ of the state to help you write the laws, as the aspect
          of the state may be influenced by other aspects of the state.
22  For example, if told to focus on Zombies, you should write laws that govern the
         behavior of Zombies. This behavior may be influenced by other parts of the
         state such as the player's actions or position.
23  If told to focus on the player, you should write laws that model how the player's
          state changes. Again, these effects may be influenced by the entities that
         the player is interacting with.
24
25  ## Guidelines for Writing Laws
26  - Some laws may be dependent on an action being taken, or a particular state of
         the world, while others may always apply. For these, the precondition can
         always be `True`.
27  - Make use of `adjacent_to_player` and `get_target_tile` to help you write laws
         about interactions between the player and other entities.
28  - Do NOT use `entity_id` when writing laws. You should instead write laws that
         apply to a type of entity, e.g. `ZombieState` or `CowState`.
29  - When modifying attributes, use RELATIVE assignments rather than absolute
         assignments. For example, instead of changing a entity's position via `
         entity.position.x = DiscreteDistribution(support=[7])`, use `entity.position
         .x = DiscreteDistribution(support=[entity.position.x + delta])`. The only
         exception to this is when modifying the materials field.
30  - Use the helper functions `get_object_of_type_in_update_range`, and `
         get_objects_in_update_range` rather than writing your own iteration logic.
31  - You DO NOT need to use the `entity_id` attribute. Use `get_target_tile` to get
         the tile or entity targeted by the player. Use `adjacent_to_player` to check
          if an entity is adjacent to the player for interactions between the player
         and other entities.
32  - Consider writing laws that make "soft" predictions. For example, if you see an
         entity moving but are unsure if it is a general principle, you can assign a
         discrete distribution to the entity's position to represent your uncertainty
         . Example: `entity.position.x = DiscreteDistribution(support=[entity.
         position.x + delta_a, entity.position.x - delta_b, ...])`.
33  - You can speculatively pose laws, but these should go last. Speculative laws are
          those that were not directly observed in the transition, but those that you
          believe might exist. For example, given that you have identified a law
         about a certain crafting recipe, you can speculatively pose a law about
         _other_ crafting recipes that you believe might exist.
34
35
36
37  ## Formatting Instructions
38  Structure your response exactly as follows. **You can write multiple laws by
         repeating the pattern below for each law:**
39
40  ```xml
41  <keyChanges>
42  List the specific, concrete changes that occurred between the observations:
```

```
43 - What entities appeared, disappeared, or moved
44 - What stats/values changed and by how much
45 - What items were added/removed from inventory
46 - Any other measurable state differences
47 </keyChanges>
48 <naturalLanguageLaw>
49 Write a clear, concise description of the game rule that explains these changes:
50 - What triggers this law (the preconditions)
51 - What the law does (the effects/transformations)
52 - Any important parameters or variations
53 - Give the law a descriptive name
54 </naturalLanguageLaw>
55 <lawCode>
56 ```python
57 class YourLawNameHere:
58     def __init__(self, param1: type = default_value, param2: type = default_value
       ):
59         """Initialize with configurable parameters."""
60         self.param1 = param1
61         self.param2 = param2
62         # Add any lookup tables or constants here
63
64     def precondition(self, current_state: WorldState, action: str) -> bool:
65         """Return True if this law should apply to the given state and action."""
66         # Implement your precondition logic here
67         # Check action type, entity presence, player state, etc.
68         return False  # Replace with actual logic
69
70     def effect(self, current_state: WorldState, action: str) -> None:
71         """Apply the law by modifying the world state."""
72         # Implement the state transformation here
73         # Modify entities, player stats, inventory, etc.
74         # Use DiscreteDistribution(support=[value]) to set deterministic
       predictions
75         # Example: current_state.player.health = DiscreteDistribution(support=[
       new_health])
76         pass  # Replace with actual implementation
77 ```
78 </lawCode>
79
80 <keyChanges>
81 [Changes for second law...]
82 </keyChanges>
83 <naturalLanguageLaw>
84 [Description of second law...]
85 </naturalLanguageLaw>
86 <lawCode>
87 ```python
88 class YourSecondLawNameHere:
89     # [Implementation of second law...]
90 ```
91 </lawCode>
92 ```
93
94 **Critical Formatting Notes**:
95 - **Write multiple laws when you observe multiple distinct changes** - each law
       should focus on a single type of change
96 - Use exactly these XML-style tags: `<keyChanges>`, `<naturalLanguageLaw>`, `<
       lawCode>`
97 - Close each tag properly: `</keyChanges>`, `</naturalLanguageLaw>`, `</lawCode>`
98 - Put all Python code inside triple backticks within the `<lawCode>` section
99 - Be precise and specific in the key changes - use exact numbers and entity names
        from the observations
```

```
100  - Make the natural language law description clear enough that another programmer
            could implement it independently
101  - Only output the code for the law, not the entire file. Assume the `WorldState`
            class as well as its components are already defined.
102  - Format your response well, with newlines between the tags and code blocks.
103  - **Each law should be completely self-contained** - repeat the full XML
            structure for each law you write.
104
105  ## WorldState
106  The world state is a Pydantic model that represents the complete game world state
            . The world laws you write will operate on this state.
107
108  ```python
109  {{ world_state_schema }}
110  ```
111
112  # World Laws
113  Each world law must conform to the following interface:
114
115  ```python
116  class WorldLaw:
117      def precondition(self, current_state: WorldState, action: str) -> bool:
118          """Return True if this law should apply to the given state and action."""
119          ...
120
121      def effect(self, current_state: WorldState, action: str) -> None:
122          """Apply the law by modifying the world state."""
123          # Use DiscreteDistribution(support=[value]) to set deterministic
        predictions
124          # Example: current_state.player.health = DiscreteDistribution(support=[
        new_health])
125          ...
126  ```
127  You may add any additional fields or methods to the class as needed.
128
129  ## DiscreteDistribution Usage
130  When modifying state values in your law's `effect` method, you must wrap the new
            values with `DiscreteDistribution`:
131
132  ```python
133  # For deterministic predictions:
134  current_state.some.value = DiscreteDistribution(support=[new_health])
135
136  # For stochastic predictions (if needed):
137  current_state.some_value = DiscreteDistribution(support=[value1, value2, value3])
138  ```
139
140  The `DiscreteDistribution` class represents probabilistic predictions over
            discrete values. For deterministic laws, you typically provide a single
            value in the support list. For stochastic laws, you provide multiple values
            in the support list to represent the possible outcomes.
141
142  When accessing the materials field, pay attention to the `MaterialT` type.
            Everything in the `materials` field is a `MaterialT`. Do not use the emojis
            in the world map, they are only there for your convenience.
143
144  # Your Turn
145  ## Aspect of the State
146  Focus on modeling changes to the following aspect of the state:
147  {{ aspect_of_state }}
148
149  ## Focused Changes for {{ aspect_of_state }}
150  {{ aspect_changes }}
```

```
151
152  ## View Legend
153  {{ view_legend }}
154
155  ## State
156  ```json
157  {{ state }}
158  ```
159  ### Local View
160  ```
161  {{ local_view }}
162  ```
163
164  ## Action
165  The action taken was: "{{ action }}"
```

**Law Generation and Parsing.** The generated prompt is sent to an LLM, which is instructed to return one or more atomic laws that explain the observed changes for the specified aspect. An atomic law is a simple, modular rule focused on a single game mechanic. The LLM's response is formatted using XML-style tags to clearly delineate the key components of each proposed law.

The expected format for a single law is:

```
<keyChanges>...</keyChanges>
<naturalLanguageLaw>...</naturalLanguageLaw>
<lawCode>
```python
class LawName:
    def precondition(self, state, action): ...
    def effect(self, state, action): ...
```

</lawCode>
```

We parse this semi-structured text to extract the natural language description and the executable Python code for each proposed law. This is done by searching for the corresponding tags and extracting their content. The Python code is then loaded as a candidate law for the subsequent parameter inference stage.

```python
def synthesize_laws_from_trajectory(trajectory: list[Transition]) -> list[Law]:
    candidate_laws = []

    # Iterate over all transitions from the exploration data
    for transition in trajectory:
        s_t, action, s_t_plus_1 = transition

        # 1. Detect which aspects of the state have changed
        changed_aspects = []
        for detector in ALL_DETECTORS:
            if detector.has_changes(s_t, s_t_plus_1):
                changed_aspects.append(detector.aspect_name())

        # 2. For each detected change, generate laws
        for aspect in changed_aspects:
            # 2a. Render a detailed prompt for the LLM
            prompt = render_synthesis_prompt(
                state=s_t,
                action=action,
                next_state=s_t_plus_1,
                aspect_of_state=aspect
            )

            # 2b. Query the LLM to synthesize laws
```

```
25            llm_response_text = call_llm(prompt)
26
27            # 2c. Parse the response to extract structured laws
28            parsed_laws = parse_laws_from_response(llm_response_text)
29            candidate_laws.extend(parsed_laws)
30
31    return candidate_laws
```

Listing 17: High-level overview of the law synthesis pipeline.

**Box G.3| Zombie Fighter Plan**

```
1  def craft_wooden_sword_plan(
2      state: WorldState,
3      transition_fn: Callable[[WorldState, CrafterAction], WorldState],
4      num_trees: int = 3
5  ) -> WorldState:
6      trees_chopped = 0
7      pathfind_option = PlayerPathfindOption(
8          lambda s: find_closest_material_of_type(s, "tree")[1]
9      )
10     interact_option = PlayerInteractAdjacentOption(
11         lambda s: find_closest_material_of_type(s, "tree")[1]
12     )
13
14     # Gather wood by iterating between pathfinding and interaction
15     while trees_chopped < num_trees:
16         try:
17             action = pathfind_option.action(state)
18         except TerminationCondition:
19             action = interact_option.action(state)
20             if action == "do":
21                 trees_chopped += 1
22         state = transition_fn(state, action)
23
24     # Place crafting table and craft sword
25     state = transition_fn(state, "place_table")
26     state = transition_fn(state, "make_wood_sword")
27
28     return state
29
30
31 def defeat_zombies_plan(
32     state: WorldState,
33     transition_fn: Callable[[WorldState, CrafterAction], WorldState],
34     zombie_ids: list[int],
35     max_steps_per_zombie: int = 10
36 ) -> WorldState:
37     for zombie_id in zombie_ids:
38         combat_option = CombatFixedEntityOption(entity_id=zombie_id)
39
40         for _ in range(max_steps_per_zombie):
41             try:
42                 action = combat_option.action(state)
43                 state = transition_fn(state, action)
44             except TerminationCondition:
45                 break  # Zombie defeated
46
47     return state
48
49
50 def sword_then_zombies_plan(
51     state: WorldState,
```

```
52     transition_fn: Callable[[WorldState, CrafterAction], WorldState],
53     zombie_ids: list[int]
54 ) -> WorldState:
55     """
56     High-level plan: Craft weapon before engaging in combat.
57     Composes two sub-plans into a complete strategy.
58     """
59     # Sub-plan 1: Obtain weapon
60     state = craft_wooden_sword_plan(state, transition_fn, num_trees=3)
61
62     # Sub-plan 2: Defeat enemies
63     state = defeat_zombies_plan(state, transition_fn, zombie_ids)
64
65     return state
```

## H  PROBABILISTIC MODELING OF PURE FUNCTIONS

In this section, we clarify the distinction between the environment's pure functional interface and the choice to learn a probabilistic world model.

We use the term "pure function" in the formal computer science sense (referential transparency and absence of side effects) (Backus, 1978), following the design patterns of modern JAX-based environments like Brax (Freeman et al., 2021) and Craftax (Matthews et al., 2024). In Crafter-OO, the transition function explicitly takes the entire state (including the PRNG state and all entity attributes) as input. This ensures referential transparency, since calling the function with the same inputs guarantees the exact same output, whereas a standard environment would diverge due to hidden state mutations.

```
1  # Standard "Impure" Environment
2  # Hidden state (e.g., rng, cooldowns) mutates inside env
3  obs1 = env.step(action)
4  obs2 = env.step(action)
5  # Result: obs1 != obs2 (The hidden state changed between calls)
6
7  # Crafter-OO "Pure" Function
8  # All state is explicit; no side effects
9  s1, rng1 = transition(state, action, rng)
10 s2, rng2 = transition(state, action, rng)
11 # Result: s1 == s2 (Identical inputs guarantee identical outputs)
```
Listing 18: Comparison of impure vs. pure environment interfaces.

Although the transition function is pure, meaning a deterministic world model is *theoretically* possible if the agent could perfectly model the evolution of the global PRNG state, such a model is difficult to learn in practice. It requires overfitting to the simulator's serial execution order, which is undesirable for several reasons.

**PRNG Scheduling.**   In a simulation with a shared global PRNG, the exact next state depends on the order in which the RNG is consumed. For example, if the simulator updates zombie_a then zombie_b, the RNG stream advances differently than if the update order were swapped. To learn a deterministic model, the agent would have to perfectly replicate the simulator's internal scheduling logic rather than what is commonly understood as a "law of the environment."

**Micro vs. Macroscopic Physics.**   We motivate this using the distinction in statistical mechanics between micro-state trajectories and macroscopic laws. While the motion of every gas molecule in classical physics is theoretically deterministic given precise initial conditions (the "micro-state"), attempting to model these trajectories is intractable and brittle, analogous to overfitting the simulator's execution trace. Instead, we aim to discover robust "macroscopic" physical laws (e.g., "zombies move randomly but chase the player when closer than 5 units"), which requires modeling the distribution of outcomes to capture the valid aleatoric uncertainty.

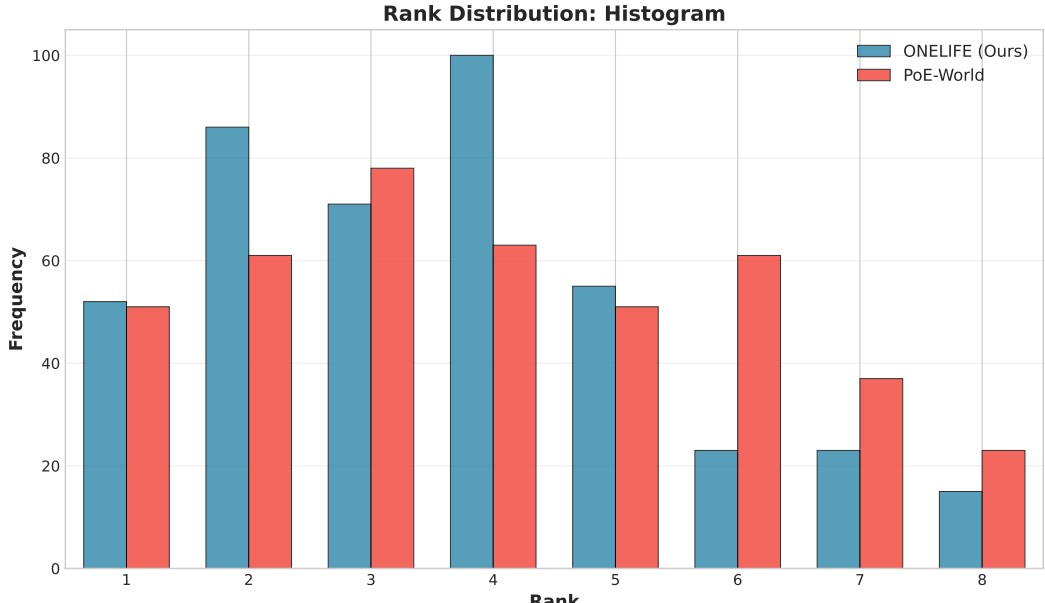

Figure 7: Distribution of raw ranks assigned by PoE-World vs ONELIFE's world models in our evaluation. Generally, ONELIFE's world model is better at assigning a high rank to the ground truth state.

**Effect on Hypothesis Verification.** This distinction determines how we validate our understanding of the world. Validating a deterministic model requires verifying the microscopic trajectory, since the predicted attribute value must exactly match the observed value. This is brittle; a law that is correct (carries out the same computations as the true transition function) may be rejected simply because the specific observed path of the RNG led to a different outcome than predicted. In contrast, a probabilistic formulation allows us to validate macroscopic laws. By evaluating the likelihood of the observation under the predicted distribution, we can confirm that a law is distributionally correct without requiring the prediction to match the specific, arbitrary path of the simulator's RNG.

## I   BASELINES

- **Random World Model:** A model that assigns a uniform probability to all candidate states in the discriminative task. Its performance is equivalent to random guessing and serves as a sanity check for discriminative accuracy.
- **WorldCoder** (Tang et al., 2024): A model-based agent that synthesizes a Python program for the transition function using an LLM. It employs an iterative refinement strategy, prompting the LLM to debug the code when it contradicts observed data. Crucially, WorldCoder assumes the environment is deterministic; we include it to evaluate how well a monolithic, deterministic program synthesis approach copes with the stochastic dynamics of Crafter-OO.
- **PoE-World** (Piriyakulkij et al., 2025): A state-of-the-art symbolic world model that scaled symbolic world modeling to domains like Atari. Both PoE-World and ONELIFE represent the transition function as a weighted product of programs, though the structure of the programs and inference algorithms differ. Because PoE-World's law synthesis component is Atari-specific and relies on online interaction using human-provided goals, we reimplement this baseline with our exploration policy and law synthesizer, noting that this makes it a stronger baseline (without these changes, PoE-World's Atari-specific implementation would be fundamentally incompatible with Crafter's state).

## J   VISUALIZING RANK DISTRIBUTIONS

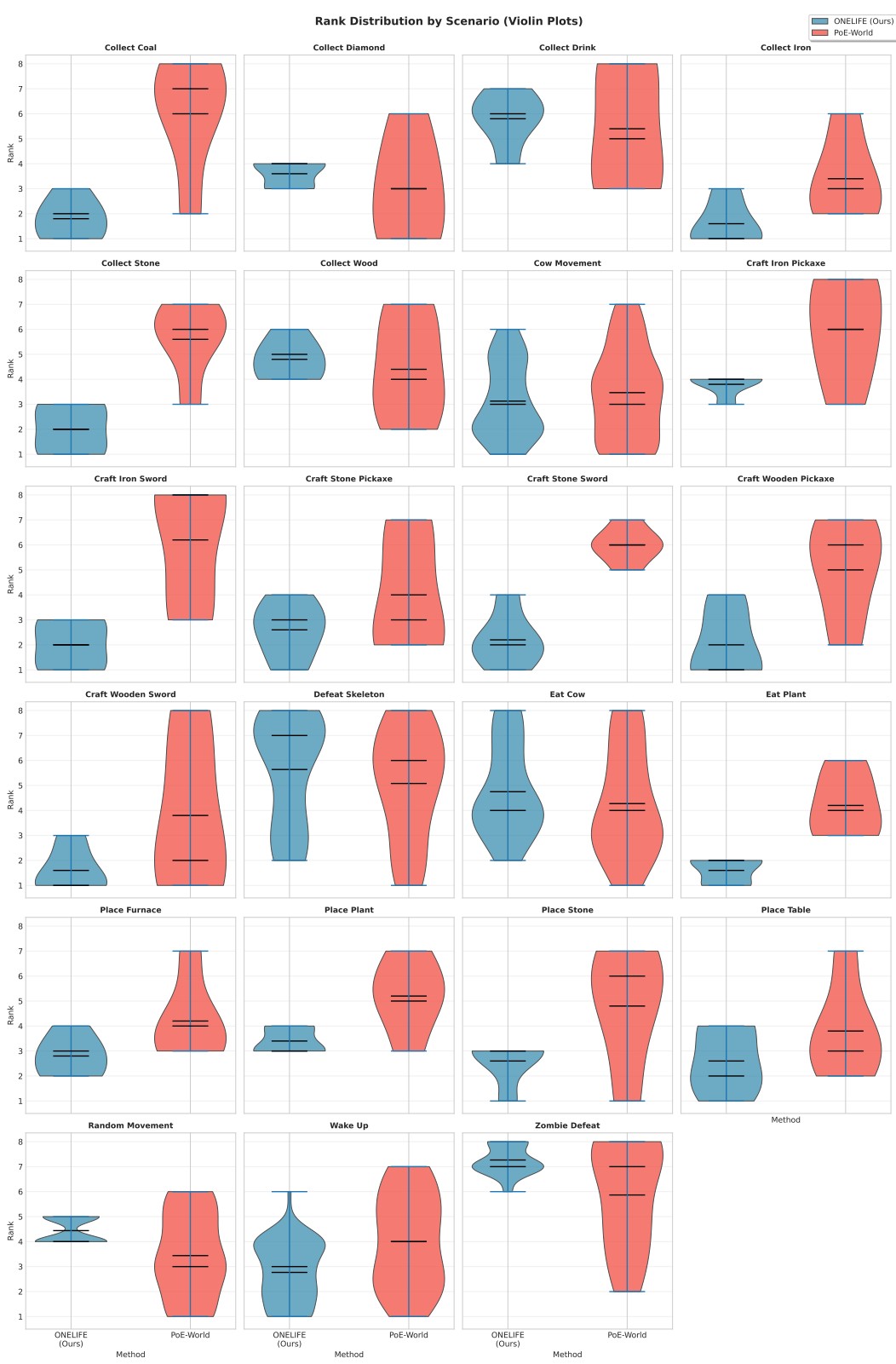

Figure 8: Distribution of raw ranks assigned by PoE-World vs ONELIFE's world models in our evaluation, broken down by scenario. Across most scenarios, ONELIFE's world model is better at choosing the ground-truth next state.

