# OpenReview forum: "One Life to Learn: Inferring Symbolic World Models for Stochastic Environments from Unguided Exploration"
_ICLR.cc/2026/Conference — ICLR 2026 Poster_

### Official Review · Reviewer_SHXj · 2025-10-17

**Soundness:** 3
**Presentation:** 2
**Contribution:** 2
**Rating:** 4
**Confidence:** 4

**Summary:**

The paper presents an approach to learning symbolic world models from observations in an MDP.

In the proposed framework, a world model is a weighted set of “laws,” each law being a pair of a precondition and an “effect” function that modifies attributes of an input state given an input action. The laws are synthesized by prompting an LLM, and, given the laws, the weights are optimized by maximizing with gradient descent the likelihood of a set of transitions.

The authors conduct experiments on a custom environment that reimplements Crafter (a popular RL environment) but exposes the entire state in text form. The experiments are conducted in a controlled (but significantly limiting) setting and compare the proposed system to a recent baseline, showing some improvement.

**Strengths:**

- Significance: the work leverages LLMs to synthesize “reactive” laws that are used to parametrize a world model. Even if I don't consider the proposed method's novelty one of its strengths, in my opinion it is of interest to the community to understand the performance (and limitations) of this kind of system, and I expect the results in this work to inform future research. Thus, **I consider significance to be one of the strengths of the paper.**
- Quality: the design choices in this framework are sensible (even if at times the motivations are not clearly articulated). Furthermore, the method appears to be relatively well-documented in the appendix, so even if the results are in some sense negative (the synthesized world models appear to be far from perfect), the experiment results are informative. Thus, **with some changes (see suggestions), I would consider the quality of the work to be adequate for this venue.**

**Weaknesses:**

The paper is generally well-written, but at times the manuscript could be more precise. In particular:
- Some aspects of the method are not described with enough detail (see suggestions 2, 3 and 5).
- Some aspects of the experimental setup should be emphasized (see suggestions 1 and 6).
- The results are a bit difficult to interpret due to the use of MRR (see suggestion 4).
- Some of the design choices are not motivated (see suggestion 7).

The experimental setup has significant limitations:
- The experiments only include one baseline (see suggestion 8).
- Experiments are performed in a single domain. I do not consider this to be a critical flaw, but it does limit the significance of the experiments.

**Questions:**

Questions:
- Why are the weights necessary? My guess is that they serve to “soften” the preconditions (which are deterministic predicates), but there is no actual motivation for the inclusion of the weights in the manuscript.
- What is the space of laws? In the Appendix it can be observed that you have a (custom?) library with which you specify the distributions of attributes for the next state, but it is never stated what space is covered with the library (e.g., what types of distributions can be represented).
- What is the experimental setup for measuring the rank? My understanding is that 10 “candidate next states” are generated with a method that leverages domain knowledge to create plausible candidates (one of which is the actual ground truth next state), and then the rank of the ground truth next state among that set is measured. Is this correct?
- Why use reciprocal rank instead of (raw) rank?
- In line 281 you state, “The policy is not provided with specific knowledge of the environment,” but in Box F.1 it can be seen that the exploration policy does leverage domain knowledge (e.g., 1588: “you may encounter creatures that are hostile and will attack you”). This must be clarified.
- Experiments are conducted per scenario, where scenarios. Does this mean that a full model of the environment is never synthesized? I.e., a model that captures all the behaviors that could happen.
- The transition function of the environment is described as a “pure transition function.” Is this accurate? If so, why not synthesize a deterministic world model?
- Have you evaluated the model predicting multiple next states in sequence? Good-enough performance in this is necessary for model-based planning and training. Based on your results (18% rank @ one), I would expect the model to perform poorly (it is still useful for the community to know that this kind of system does not perform well enough).
- Why did you not include other systems as baselines (e.g., WorldCoder)? Your baseline is adequate, but a single data point to compare to is not ideal.
- Is the core difference between the proposed system and PoE that the proposed system aims to synthesize “atomic” laws?

Suggestions:
1. You must correctly state that the exploration policy leverages domain knowledge to get meaningful trajectories (Box F.1). This is important because the quality of the trajectories influences the synthesis of the world model.
2. I suggest fully reworking the description of the synthesizer (line 288), as it cannot be inferred at all that the synthesizer is a custom routine that calls an LLM to synthesize a law for each detected change in the state (Listing 17).
3. I suggest you describe (with some level of precision) the set of laws that can be synthesized. In the Appendix it is shown to be Python programs that use a library to describe the distributions of each attribute of the state in the next state. This should be stated in the main text, as it informs the reader about the level of expressivity of the system and the types of programs that are being synthesized, both of which are important to understand the work.
4. I suggest you provide a visualization of the ranks of the states (e.g., a scatter plot with a corresponding histogram). This would help the reader better understand the performance of the model (and baseline ideally).
5. The laws are stochastic (as can be seen in, e.g., Box A.4), but in line 237 you use notation typically reserved for (deterministic) functions. I suggest making the definition of all parts of the world model mathematically precise.
6. If the per-scenario experimental setup implies that a “full” world model (i.e., a model that covers the entire state space) is never synthesized, then this should be stated in the manuscript, as it helps the reader understand the scope of the experiments.
7. Motivate the use of the law's weights. Ideally this is done with both a conceptual explanation and an ablation study.
8. Adding another baseline would significantly improve the experimental section.

---

> ### Author Response · Authors · 2025-11-21
> **Response to SHXj (Part 1)**
>
> Thank you for the detailed and thoughtful review\! We are encouraged that you consider the '**significance to be one of the strengths of the paper**' and expect our results to **'inform future research**'. We are grateful for your detailed and constructive suggestions, which have improved the clarity and rigor of our work. We have **addressed each of your points in the revised manuscript**, including adding the WorldCoder baseline, visualizing the rank metrics, and refining the method description.
>
> ## W1: Additional detail on aspects of method (synthesizer, probability spaces, notation)
>
> > Some aspects of the method are not described with enough detail (see suggestions 2, 3 and 5).
> We have expanded the method description as requested. Specifically, we have:
>
> * **Reworked the synthesizer description** to clearly explain the routine and the LLM's specific role **(S2)**.
> * **Defined the space of laws** to clarify the system's expressivity **(S3)**.
> * **Revised mathematical notation** to better represent stochastic functions **(S5)**.
>
> ## W2: Emphasizing aspects of experimental setup (exploration policy, world model scope)
>
> > Some aspects of the experimental setup should be emphasized (see suggestions 1 and 6).
>
> We have updated the manuscript to emphasize these setup details:
>
> * **Exploration Policy:** We now explicitly state that the policy leverages general domain knowledge to obtain meaningful trajectories **(S1)**.
> * **Model Scope:** We clarified that we synthesize a single, compositional world model utilized across all scenarios, rather than separate models per scenario **(S6)**.
>
> ## W3: Visualization of raw ranks
>
> > The results are a bit difficult to interpret due to the use of MRR (see suggestion 4).
>
> **Visualization:** To aid interpretation, we have added a scatter plot and histogram of the raw ranks alongside the MRR metrics and motivated the use of MRR **(S4)**.
>
> ##  W4: Motivating and ablating learnable weights
>
> >  Some of the design choices are not motivated (see suggestion 7).
>
> **Motivation for Weights:** We added a section explicitly motivating the learnable weights, explaining their necessity for **pruning** incorrect hypotheses and **merging** predictions in a stochastic environment **(S7),** as well as an **ablation** **without weight learning (L445 in Table 1 – “No Inference”)**.
>
>
>
> ## W5: Adding Worldcoder as a baseline
>
> > The experiments only include one baseline (see suggestion 8).
>
> * **New Baseline:** We have added **WorldCoder** as a second baseline to provide a more robust comparison **(S8)**.
>
> | Method | Rank @ 1 ↑ | MRR ↑ | Raw Edit Dist. ↓ | Norm. Edit Dist. ↓ |
> | :---- | :---: | :---: | :---: | :---: |
> | WorldCoder | 0.0% | 0.264 | 27.180 | 0.181 |
> | PoE-World | 10.8% | 0.351 | 10.634 | 0.071 |
> | OneLife | **18.7%** | **0.479** | **8.764** | **0.058** |
>
> ## W6: Clarifying Engineering Scale and Rigor of the Crafter-OO Environment
>
> > W6: Experiments are performed in a single domain. I do not consider this to be a critical flaw, but it does limit the significance of the experiments.
>
> We do only evaluate on Crafter-OO, but it features a wide range of mechanics, is stochastic, and significant manual effort went into designing a rigorous evaluation. Crafter-OO, like Crafter, is highly versatile and can be thought of as a meta-domain upon which to implement a variety of diverse tasks. Additionally, implementing Crafter-OO was a multi-month engineering effort. We do not think it would be possible to add another environment of equal complexity and an evaluation of equal rigor within the timeframe of the rebuttal. We provide a link to the anonymous version of the environment’s source code (https://anonymous.4open.science/r/crafter-oo-iclr26-anon-35C2) — it totals over 4426 lines of code over the original Crafter.

---

> ### Author Response · Authors · 2025-11-21
> **Response to SHXj (Part 2)**
>
> ## Q1: Why are the weights necessary?
>
> > Why are the weights necessary? My guess is that they serve to “soften” the preconditions (which are deterministic predicates), but there is no actual motivation for the inclusion of the weights in the manuscript.
>
> The learnable weights are necessary for model selection over the set of synthesized candidate laws. Our synthesizer (Sec. 3.3) is designed to propose a large set of simple, atomic laws to explain observed transitions. This process inevitably produces hypotheses that are incorrect, redundant, or only partially capture a stochastic phenomenon. The weights provide the mechanism to adjudicate between these competing laws based on their predictive accuracy.
>
> Specifically, the weights serve two primary functions:
>
> 1. **Learning and Pruning:** During optimization, laws that consistently fail to predict observed outcomes will have their corresponding weights driven towards zero. This effectively prunes them from the world model. This allows our framework to adopt a "propose-and-prune" strategy, where the synthesizer can be over-generative and the inference algorithm automatically filters out incorrect hypotheses.
> 2. **Merging Predictions:** In a stochastic environment, it can be difficult to distinguish between a law that is fundamentally incorrect versus one that is distributionally correct but whose single sample prediction did not match a specific outcome. The weights allow multiple plausible laws to contribute to the final predictive distribution for an observable (Equation 1), with each law's "vote" scaled by its learned weight.
>
> **We have added a paragraph (L291-295)** to explicitly state this motivation, clarifying that the weights facilitate a form of continuous model selection and hypothesis merging within our probabilistic programming framework. Additionally, Table 1 ablates the effect of removing the weights:
>
> | Method | Rank @ 1 ↑ | MRR ↑ | Raw Edit Dist. ↓ | Norm. Edit Dist. ↓ |
> | :---- | :---: | :---: | :---: | :---: |
> | OneLife (No Inference) | 13.0% | 0.429 | 8.540 | 0.057 |
> | OneLife (With Inference) | 18.7% | 0.479 | 8.764 | 0.058 |
> | Difference | \+5.7% | \+0.050 | \+0.224 | \+0.001 |
>
> ## Q2: What is the space of laws?
>
> > What is the space of laws? In the Appendix it can be observed that you have a (custom?) library with which you specify the distributions of attributes for the next state, but it is never stated what space is covered with the library (e.g., what types of distributions can be represented).
>
> The library covers categorical distributions and discrete distributions. In principle, this could be extended to also cover continuous distributions — our inference algorithm works as long as it is able to query the likelihood of an observed data point. **We have added this information to L275-277.**
>
> ## Q3: What is the experimental setup for measuring the rank?
>
> > What is the experimental setup for measuring the rank? My understanding is that 10 “candidate next states” are generated with a method that leverages domain knowledge to create plausible candidates (one of which is the actual ground truth next state), and then the rank of the ground truth next state among that set is measured. Is this correct?
>
> You are correct that we use domain knowledge to generate plausible "distractor" candidates and rank the ground truth among them. We generate the ground truth and the distractors as follows:
>
> **1\. Ground Truth Generation (Scenarios)** To obtain diverse ground truth transitions for _evaluation_  (described in **Section 4.1** and **Appendix D/E**), we employ a suite of 30+ scenarios that cover every achievement in the Crafter tech tree:
>
> * These range from basic mechanics (e.g., `collect_wood`, `place_table`) to complex interactions (e.g., `craft_iron_sword`, `defeat_skeleton`).
> * Each scenario starts from specific initial state and a scripted policy designed to reliably trigger that specific mechanic. Running the policy produces transitions.
>
> **2\. Candidate Generation (Mutators)** To create the distractors, we use a bank of mutators (Section 4.1, Appendix D).
>
> * These function as "bad laws". You can think of them as programs that apply semantically meaningful but impermissible (under the true transition function) changes.
> * Since mutators have precondition checks (e.g., combat mutators only apply during combat), the exact number of candidates varies per transition.
> * We guarantee a minimum of **7** candidates (distractors \+ ground truth), with a typical range of **7 to 11** candidates per transition.
>
> **We updated L471-474** to explicitly state the range of candidate states ($N \\in \[7, 11\]$) to clarify this.

---

> ### Author Response · Authors · 2025-11-21
> **Response to SHXj (Part 3)**
>
> ## Q4: Why use reciprocal rank instead of raw rank?
>
> > Why use reciprocal rank instead of (raw) rank?
>
> We selected Mean Reciprocal Rank (MRR) over Mean Rank because it offers better alignment with downstream planning utility and greater metric stability. Both MRR and Mean Rank yield the same ordering over methods.
>
> **Alignment with Simulation Utility**: Forward simulation relies on consistently sampling the correct next state to maintain valid causal chains. MRR's inverse scaling ($1/r$) imposes a steep penalty for missing the top rank (e.g., $1.0 \\to 0.5$), accurately reflecting that even a near miss (Rank 2\) often represents a physically invalid state (e.g., teleportation) that breaks the simulation. Mean Rank treats all ranks equally.
>
> **Metric Standardization and Independence**: Raw Mean Rank is uninterpretable without knowing the candidate set size ($N$), as the baseline for random guessing scales linearly with $N$ (expected rank $\\approx N/2$). For example, a Mean Rank of 5 represents poor performance in a pool of 10 items (barely better than random) but exceptional performance in a pool of 1,000 items. This standardization is useful for our evaluation, since the number of valid distractors naturally varies depending on how many mutators are applicable to a given state-action pair. MRR provides a bounded score on $\[0,1\]$ that remains invariant to these fluctuations; an MRR of 0.2 consistently means that the true state appeared at rank 5, regardless of the varying candidate pool size.
>
> **We’ve added justification of the choice of MRR in L431-451 in addition to request visualization (Appendix J; Fig7, Fig8)**.
>
> ## Q5: What domain knowledge does the exploration policy use?
>
> > In line 281 you state, “The policy is not provided with specific knowledge of the environment,” but in Box F.1 it can be seen that the exploration policy does leverage domain knowledge (e.g., 1588: “you may encounter creatures that are hostile and will attack you”). This must be clarified.
>
> Thank you for highlighting this ambiguity. We distinguish between **general genre priors** — high-level concepts common to the *class* of open-world survival environments (e.g., the existence of hostile entities, ability to collect resources, or craft tools) — and **environment-specific dynamics** (e.g., specific facts like "Zombies chase players" or "Wood is needed to make a pickaxe").
>
> The exploration policy is provided with the former to prevent aimless behavior typical of random policies, but it is strictly withheld from the latter. This design mimics a realistic scenario where an agent enters a new environment possessing broad intuition about the genre, but must reverse-engineer the specific laws and mechanics of that unique world from scratch.
>
> **We have added the above text clarifying this in L320-327**.
>
> ## Q6: Is a full model of the environment synthesized?
>
> > Experiments are conducted per scenario, where scenarios. Does this mean that a full model of the environment is never synthesized? I.e., a model that captures all the behaviors that could happen.
>
> To clarify, we do not synthesize separate models for each scenario. Rather, we synthesize **one single compositional world model** from the unguided exploration data, and then we use the various scenarios to evaluate the specific capabilities of that one model.
>
> To elaborate on the process:
>
> 1. **Model Synthesis:** As described in Section 3.3, ONELIFE synthesizes a single world model (a mixture of symbolic laws) from a single episode of unguided exploration. This exploration policy is not aware of specific scenarios; its goal is simply to uncover as many mechanics as possible. The result is one global set of programmatic laws intended to capture the dynamics of the entire environment.
> 2. **Per-Scenario Evaluation:** The "scenarios" described in Section 4.1 are used exclusively for testing, not training. Because the environment is complex, a global accuracy metric might be dominated by common transitions (like moving) while masking failures in rare but critical mechanics (like combat or crafting). Therefore, we evaluate the *same* single world model across 23 different scenarios to inspect its knowledge of specific game mechanics.
>
> The results represented in Fig 4 show the performance of one unified ONELIFE world model being tested in many scenarios. A world model that scores perfectly across all these scenarios would effectively be a complete model of the environment's behaviors.

---

> ### Author Response · Authors · 2025-11-21
> **Response to SHXj (Part 4)**
>
> ## Q7: If the transition function is pure, why can’t we synthesize a deterministic world model?
>
> > The transition function of the environment is described as a “pure transition function.” Is this accurate? If so, why not synthesize a deterministic world model?
>
> **Part A: Is the transition function a pure function?**
> We use the term "pure function" in the sense of   (having referential transparency and absence of side effects)\[1\], following the design patterns of modern JAX-based environments like Brax \[2\] and Craftax \[3\]. In Crafter-OO, the transition function explicitly takes the entire state (including the PRNG state and all entity attributes) as input. This ensures referential transparency, since calling the function with the same inputs guarantees the exact same output, whereas a standard environment would diverge due to hidden state mutations.
>
> ```
> # Standard "Impure" Environment
> # Hidden state (e.g., rng, cooldowns) mutates inside env
> obs1 = env.step(action)
> obs2 = env.step(action)
> # Result: obs1 != obs2 (The hidden state changed between calls)
>
> # Crafter-OO "Pure" Function
> # All state is explicit; no side effects
> s1, rng1 = transition(state, action, rng)
> s2, rng2 = transition(state, action, rng)
> # Result: s1 == s2 (Identical inputs guarantee identical outputs)
> ```
>
> **Part B: If so, why learn a probabilistic world model?**
> It is true that because the transition function is pure, learning a deterministic world model is *theoretically* possible if the agent could perfectly model the evolution of the global PRNG state. However, in practice, this is very difficult to learn and undesirable because it requires overfitting to the simulator's serial execution order.
>
> **PRNG Scheduling** In a simulation with a shared global PRNG, the exact next state depends on the order in which the RNG is consumed. For example, if the simulator updates `zombie_a` then `zombie_b`, the RNG stream advances differently than if it updated `zombie_b`, then `skeleton_a`, then `zombie_a`. To learn a deterministic model, the agent would have to perfectly replicate the simulator's internal scheduling logic, rather than what is commonly understood as a “law of the environment”, which we argue in more detail below.
>
> **Micro vs Macroscopic Physics** We can motivate this using the distinction in statistical mechanics between micro-state trajectories and macroscopic laws. While the motion of every gas molecule in classical physics is theoretically deterministic given precise initial conditions (the "micro-state"), attempting to model these trajectories is intractable and brittle, analogous to overfitting the simulator's execution trace. Instead, we aim to discover the robust "macroscopic" physical laws (e.g., "zombies move randomly but chase the player when $L^1$ to player $\<5$"), which requires modeling the distribution of outcomes to capture the valid aleatoric uncertainty.
>
> **Effect on Hypothesis Verification** This distinction determines how we validate our understanding of the world. Validating a deterministic model requires verifying the microscopic trajectory, since the predicted attribute value must exactly match the observed value. This is brittle. For example a law that is “correct” (carries out the same computations as the true transition function) may be rejected simply because the specific observed path of the RNG led to a different outcome than the law predicted. In contrast, a probabilistic formulation allows us to validate macroscopic laws. By evaluating the likelihood of the observation under the predicted distribution, we can confirm that a law is distributionally correct without requiring the prediction to match the specific, arbitrary path of the simulator's RNG.
>
> **We’ve added a discussion of this in Appendix H (L2168-2215) and a pointer in the main text L196-198 pointing to it.**
>
> **References:**
>
> \[1\] Backus, John. "Can programming be liberated from the von Neumann style? A functional style and its algebra of programs." *Communications of the ACM* 21.8 (1978): 613-641.
>
> \[2\] Freeman et al. "Brax: A Differentiable Physics Engine for Large Scale Rigid Body Simulation." (2021).
>
> \[3\] Matthews et al. "Craftax: A Lightning-Fast Benchmark for Open-Ended Reinforcement Learning." (2024).

---

> ### Author Response · Authors · 2025-11-21
> **Response to SHXj (Part 5)**
>
> ## Q8: Have you evaluated the world model on multi-step prediction tasks?
>
> > Have you evaluated the model predicting multiple next states in sequence? Good-enough performance in this is necessary for model-based planning and training. Based on your results (18% rank @ one), I would expect the model to perform poorly (it is still useful for the community to know that this kind of system does not perform well enough).
>
> Thank you for this great suggestion\! We agree that good enough performance in multi-step simulation is necessary for model-based planning and training. **We have added a new section to the paper (Appendix A) containing experiments to address this.**
>
> To assess the practical utility of the learned world model, we evaluate its effectiveness in a planning context. Our protocol tests the model's ability to distinguish between effective and ineffective plans through forward simulation. For a set of scenarios, we define a reward function and two distinct, multi-step programmatic policies (plans) to achieve a goal. We execute rollouts of both plans entirely within our learned world model and, separately, within the ground-truth environment. The measure of success is whether the world model's simulation yields the same preference ranking over the two plans as the true environment, based on the final reward.
>
> Simulating the results of these plans tests the ability of the world model to accurately model the consequences of long sequences of actions upon the world. On average, a plan requires ≈ 18 steps to execute, with the longest plans taking \> 30 steps. The model would not be able to correctly rank these plans if it were not capturing the important causal dynamics of the environment.
>
> Our results, presented in the table below, show that across all three scenarios our learned world model correctly predicts the more effective plan. The ranking of plans generated by simulating rollouts in ONELIFE matches the ranking from the ground-truth environment. This demonstrates that ONELIFE captures a sufficiently accurate causal model of the world to support basic, goal-oriented planning.
>
> **Tables: Planning via forward simulation.** Our learned world model is used to compare alternative plans in three scenarios. This is done by executing the plans in the world model, and measuring the reward obtained by each plan. In each case, ONELIFE produces the same ranking over plans as the ground-truth environment, demonstrating its ability to capture causally relevant dynamics for goal-directed decision-making and accurately simulate long action sequences of \> 30 steps. Each plan was executed 10 times.
>
> **Scenario: Zombie Fighter** *Reward Function: Damage Per Second*
>
> | Plan Description | Avg. Steps | True Env. Reward | Preferred (True Env) | ONELIFE's WM Reward | Preferred (OneLife) |
> | :---- | :---: | :---: | :---: | :---: | :---: |
> | Harvest Wood → Craft Table → Craft Sword → Fight | 33 | 2.0 | ✓ | 2.03 | ✓ |
> | Fight Immediately | 17 | 1.0 |  | 1.67 |  |
>
> **Scenario: Stone Miner** *Reward Function: Stone Collected*
>
> | Plan Description | Avg. Steps | True Env. Reward | Preferred (True Env) | ONELIFE's WM Reward | Preferred (OneLife) |
> | :---- | :---: | :---: | :---: | :---: | :---: |
> | Harvest Wood → Craft Table → Craft Pickaxe → Mine | 31 | 3.0 | ✓ | 3.0 | ✓ |
> | Mine Immediately | 13 | 0.0 |  | 0.0 |  |
>
> **Scenario: Sword Maker** *Reward Function: Swords Crafted*
>
> | Plan Description | Avg. Steps | True Env. Reward | Preferred (True Env) | ONELIFE's WM Reward | Preferred (OneLife) |
> | :---- | :---: | :---: | :---: | :---: | :---: |
> | Reuse Crafting Table for all Swords | 5 | 4.0 | ✓ | 4.0 | ✓ |
> | Place New Table per Sword | 10 | 2.0 |  | 2.0 |  |
>
> Regarding your observation on the 18% Rank@1 score: Our analysis suggests that the low global score does not preclude utility in specific planning problems / goals due to the hierarchical structure of the environment:
>
> 1. The Rank@1 metric is averaged over the entire evaluation set, which covers the full "tech tree," including rare, late-game mechanics that the unguided agent rarely encounters or mutations.
> 2. The planning scenarios we crafted rely on foundational mechanics (e.g., movement, wood harvesting, basic crafting) that occur early in the hierarchy. The model encounters these frequently during exploration and models them with higher fidelity than the global average suggests.
> 3. Consequently, the model can successfully simulate long-horizon plans that utilize these well-learned subsystems, even while the global metric is dragged down by errors on rarer, more complex transitions or mutations that the world model has not learned to discriminate against.

---

> ### Author Response · Authors · 2025-11-21
> **Response to SHXj (Part 6)**
>
> ## Q9: Can you add another baseline?
>
> > Why did you not include other systems as baselines (e.g., WorldCoder)? Your baseline is adequate, but a single data point to compare to is not ideal.
>
> **We’ve added WorldCoder as a baseline (Table 1)**. Both PoE-World and OneLife are able to model the environment better than WorldCoder, likely because they make less restrictive assumptions (environment need not be deterministic) and do not attempt to write a single, monolithic program for the transition function (which is a challenging software engineering problem).
>
> ## Q10: Is the core difference between the proposed system and PoE that the proposed system aims to synthesize “atomic” laws?
>
> > Is the core difference between the proposed system and PoE that the proposed system aims to synthesize “atomic” laws?
>
> Yes. The main difference between our representation \+ inference algorithm and PoE-World’s is that we learn a mixture of *atomic* *laws* that each individually predict a *minimal subset of the next state*, whereas in PoE-World, each of their programmatic experts produces a prediction for the *entire next state*. Thus, we aim to ***factorize the transition function into atomic, predictive laws***, whereas PoE-World effectively learns a *superposition of world modeling expert programs*  — this causes attribute predictions from PoE-World for complex states to tend towards randomness due to contributions from irrelevant experts.
>
> Our world model is capable of making predictions about arbitrary attributes of the world, such as NPC attributes, tiles in the game map, or items in a player’s inventory. PoE-World focuses only on modeling a few selected physics-based attributes such as position, velocity, and health. This allows our inferred laws to model a diverse range of game mechanics, from movement to crafting to NPC behavior as shown in \<examples from appendix\>.
>
> **We’ve added lines clarifying this in L296-306 of the methods section.**
>
> # Suggestions
>
> > S1: You must correctly state that the exploration policy leverages domain knowledge to get meaningful trajectories (Box F.1). This is important because the quality of the trajectories influences the synthesis of the world model.
>
> **We’ve added text to this effect in L320-327.**, details are in response to Q4 above.
>
> > S2: I suggest fully reworking the description of the synthesizer (line 288), as it cannot be inferred at all that the synthesizer is a custom routine that calls an LLM to synthesize a law for each detected change in the state (Listing 17).
>
> **We’ve completely reworked our description of the synthesizer in L333-344** to walk through the concrete implementation, including the change detection and LLM generation.
>
> > S3: I suggest you describe (with some level of precision) the set of laws that can be synthesized. In the Appendix it is shown to be Python programs that use a library to describe the distributions of each attribute of the state in the next state. This should be stated in the main text, as it informs the reader about the level of expressivity of the system and the types of programs that are being synthesized, both of which are important to understand the work.
>
> **We’ve added a description of the set of laws that can be synthesized in L275-277**, details are in response to Q2 above.
>
> > S4: I suggest you provide a visualization of the ranks of the states (e.g., a scatter plot with a corresponding histogram). This would help the reader better understand the performance of the model (and baseline ideally).
>
> **We’ve added these visualization in Fig 7 and Fig 8 in Appendix J**.
>
> > S5: The laws are stochastic (as can be seen in, e.g., Box A.4), but in line 237 you use notation typically reserved for (deterministic) functions. I suggest making the definition of all parts of the world model mathematically precise.
>
> **We’ve revised our notation where it was inconsistent in S3.2, making changes to L252-255 and L272-275.** It now indicates that the laws map to distributions over attributes of the next state using simplex notation $\\Delta$ or explicitly writing out the distribution.
>
> > S6: If the per-scenario experimental setup implies that a “full” world model (i.e., a model that covers the entire state space) is never synthesized, then this should be stated in the manuscript, as it helps the reader understand the scope of the experiments.
>
> We answer this in our response to Q6 above.
>
> > S7: Motivate the use of the law's weights. Ideally this is done with both a conceptual explanation and an ablation study.
>
> **We’ve added text motivating the law to L291-295**, details are in our response to Q1. **Table 1 includes a row that ablates** the performance of the world model without weight fitting.
>
> > S8: Adding another baseline would significantly improve the experimental section.
>
> **We’ve added WorldCoder as a baseline**, details are in our response to Q9.

---

> ### Comment · Reviewer_SHXj · 2025-11-25
>
> I appreciate the thoroughness of your rebuttal, which meaningfully addresses most of my concerns. I have reviewed the updated manuscript and revised my score.
>
> While there are some important limitations in the experimental setup and proposed method (that would be challenging to address in a single review cycle), I do believe the work is firmly within an acceptable level of quality for publication at ICLR.

---

> > ### Author Response · Authors · 2025-11-25
> >
> > We are thrilled by your response and the decision to raise the score to an 8! We really appreciate your statement that the work is firmly within the quality standards for the conference.  Your feedback has made the paper stronger. Thank you! :)

---

### Official Review · Reviewer_NpK1 · 2025-10-29

**Soundness:** 3
**Presentation:** 3
**Contribution:** 3
**Rating:** 6
**Confidence:** 3

**Summary:**

This paper learns a symbolic/programmatic world model for Crafter. It uses LLM for exploration to collect data for training world models. It then asks LLMs to generate codes as world models given the collected data. The programs are modeled as a probabilistic composition of simpler programs/laws, taking advantage of the decompositionability of states to speed up program synthesis. It also builds a Crafter-OO environment together with some evaluation metrics.

**Strengths:**

The new Crafter-OO environment is interesting. It's more stochastic (like the random movements of zombies) than the previous environments used in the field. It's good to see that symbolic world models also work in the Crafter environment.

The paper is generally well-written.

**Weaknesses:**

It's hard to tell the differences between program synthesizers in this paper versus PoE-World, and why the differences are important.

Although it is informative to check if the next states are predicted correctly, it would be much better to use metrics that involve long trajectories and goals, such as the success rates of solving a problem in Crafter using the learned world model.

**Questions:**

* What are the differences between the program synthesizers in this paper versus PoE-World, and why are the differences important?

* Are there metrics involving long trajectories and goals? How good are the learned world models from that perspective?

---

> ### Author Response · Authors · 2025-11-21
> **Response to reviewer NpK1 (Part 1)**
>
> Thank you for the positive appraisal of our work\! We’re glad you **found the Crafter-OO environment “interesting”** and **were pleased “that symbolic world models also work in the Crafter environment”**. We genuinely appreciate your suggestion to evaluate the model **using metrics that involve long trajectories and goals,** as it identifies a key opportunity to demonstrate the model's practical utility. We have substantially **updated the paper to include a new section** (Appendix A; L676-767) evaluating the success rates of the model in solving multi-step problems.
>
> ## W1: Explaining Importance of Synthesizer Differences in OneLife vs PoE-World
>
> > What are the differences between the program synthesizers in this paper versus PoE-World, and why are the differences important?
>
> 1. The synthesizers used in PoE-World bake in a significant amount of human knowledge about the specific mechanics in the game, their inputs, outputs, and their form. Specifically, **PoE-World has over 30 synthesizers, each with a hand-written human prompt guiding the synthesizer to generate code for a given mechanic.**
> 2. In contrast, we have a single synthesizer that identifies mechanics on the fly and writes code for the laws representing them, without guidance about what types of mechanics are in the game, their inputs, or outputs. **This is important because in an unknown environment, the synthesizer must be capable of identifying mechanics without human aid.**
> 3. Additionally, PoE-World’s synthesizers are limited to synthesizing code only about a limited set of attributes (position, velocity, health) and a limited set of objects (moving game entities). In contrast, our synthesizers consume the entire game state in a general purpose format (JSON) and can write code about any aspect of the game world, including map tiles, entities, and player inventories, to give an example. **This is important because the mechanics in Crafter-OO encompass a wide variety of state attributes and the synthesizer must be able to reason about all of them**.
>
> We have added this distinction in L345-351.

---

> ### Author Response · Authors · 2025-11-21
> **Response to reviewer NpK1 (Part 2)**
>
> ## W2: Evaluating World Models on Longer Trajectories / Multi-step Tasks
>
> > Although it is informative to check if the next states are predicted correctly, it would be much better to use metrics that involve long trajectories and goals, such as the success rates of solving a problem in Crafter using the learned world model.
>
> This is an excellent suggestion. We agree that demonstrating the utility of the learned world model for planning in goal-oriented tasks with long trajectories is a crucial evaluation. **We have added a new section to the paper containing experiments to address this (Appendix A; L676-767), and the results are summarized as follows:**
>
> To assess the practical utility of the learned world model, we evaluate its effectiveness in a planning context. Our protocol tests the model's ability to distinguish between effective and ineffective plans through forward simulation. For a set of scenarios, we define a reward function and two distinct, multi-step programmatic policies (plans) to achieve a goal. We execute rollouts of both plans entirely within our learned world model and, separately, within the ground-truth environment. The measure of success is whether the world model's simulation yields the same preference ranking over the two plans as the true environment, based on the final reward.
>
> Simulating the results of these plans tests the ability of the world model to accurately model the consequences of longer sequences of actions upon the world. On average, a plan requires ≈ 18 steps to execute, with the longest plans taking \> 30 steps. The model would not be able to correctly rank these plans if it were not capturing the important causal dynamics of the environment.
>
> Our results, presented in the table below, show that across all three scenarios our learned world model correctly predicts the more effective plan. The ranking of plans generated by simulating rollouts in ONELIFE matches the ranking from the ground-truth environment. This demonstrates that ONELIFE captures a sufficiently accurate causal model of the world to support basic, goal-oriented planning.
>
> **Tables: Planning via forward simulation.** Our learned world model is used to compare alternative plans in three scenarios. This is done by executing the plans in the world model, and measuring the reward obtained by each plan. In each case, ONELIFE produces the same ranking over plans as the ground-truth environment, demonstrating its ability to capture causally relevant dynamics for goal-directed decision-making and accurately simulate long action sequences of \> 30 steps. Each plan was executed 10 times.
>
> **Scenario: Zombie Fighter** *Reward Function: Damage Per Second*
>
> | Plan Description | Avg. Steps | True Env. Reward | Preferred (True Env) | ONELIFE's WM Reward | Preferred (OneLife) |
> | :---- | :---: | :---: | :---: | :---: | :---: |
> | Harvest Wood → Craft Table → Craft Sword → Fight | 33 | 2.0 | ✓ | 2.03 | ✓ |
> | Fight Immediately | 17 | 1.0 |  | 1.67 |  |
>
> **Scenario: Stone Miner** *Reward Function: Stone Collected*
>
> | Plan Description | Avg. Steps | True Env. Reward | Preferred (True Env) | ONELIFE's WM Reward | Preferred (OneLife) |
> | :---- | :---: | :---: | :---: | :---: | :---: |
> | Harvest Wood → Craft Table → Craft Pickaxe → Mine | 31 | 3.0 | ✓ | 3.0 | ✓ |
> | Mine Immediately | 13 | 0.0 |  | 0.0 |  |
>
> **Scenario: Sword Maker** *Reward Function: Swords Crafted*
>
> | Plan Description | Avg. Steps | True Env. Reward | Preferred (True Env) | ONELIFE's WM Reward | Preferred (OneLife) |
> | :---- | :---: | :---: | :---: | :---: | :---: |
> | Reuse Crafting Table for all Swords | 5 | 4.0 | ✓ | 4.0 | ✓ |
> | Place New Table per Sword | 10 | 2.0 |  | 2.0 |  |

---

> ### Author Response · Authors · 2025-11-26
> **Follow up to reviewer NpK1**
>
> Hi NpK1\! With only a few days left in the discussion period, we wanted to gently check in whether our rebuttal addressed all your questions. We are happy to address any remaining questions. We’ve revised the manuscript to **distinguish between the multiple handcrafted PoE-World synthesizers and our general synthesizer,** and conducted **new experiments on solving longer horizon multi-step problems in Crafter** using the learned world model\! We hope that these additional results and answers will allow you to revisit your score — otherwise, we are happy to engage further\!

---

### Official Review · Reviewer_CLdf · 2025-10-31

**Soundness:** 2
**Presentation:** 4
**Contribution:** 3
**Rating:** 4
**Confidence:** 4

**Summary:**

This paper introduces ONELIFE, a framework for learning a symbolic world model from a single, unguided "one life" exploration in a complex and stochastic environment. The core of ONELIFE is to model the world's dynamics as a "mixture of laws", which are programmatic rules with preconditions and effects. The system uses a probabilistic programming approach to infer the importance of these laws, efficiently routing credit only to the rules that are relevant to observed changes in the state. To test this method, the authors also developed Crafter-OO, a new, complex testbed based on the Crafter environment, which exposes a fully symbolic, object-oriented state. The authors demonstrate that ONELIFE is superior to a baseline in its ability to rank plausible future states, suggesting it learns a more accurate model of the environment's rules.

**Strengths:**

- The introduction of Crafter-OO is a valuable contribution to the research community, offering a new, complex testbed for agents that must operate in dynamic environments with a mixture of underlying laws.
- The paper tackles a challenging and important problem setting: learning a world's rules from minimal, unguided interaction in a hostile, stochastic environment.

**Weaknesses:**

- The setup, while realistic in its limited interaction budget, makes several unrealistic assumptions. It relies on full, symbolic observability of a structured state, which is rare. The "one life" constraint also feels artificial for many real-world tasks (e.g., robot manipulation) where mistakes are often recoverable.
- The technical novelty of the method itself is somewhat unclear. Domain inference, state tracking, and programmatic models are not new. The paper could do a better job of positioning its technical novelty against this prior work, beyond just the Crafter-OO and "one life" setting.
- The paper is unclear about the environment's specifics. For instance, it's not immediately clear why the object-oriented state isn't just described as a PDDL-based representation. A clear list or description of the agent's action space is also missing, making it hard to grasp the task's complexity.
- The framework is only evaluated on Crafter-OO. While this is a new and complex environment, testing on only one domain makes it difficult to assess the generalizability of the ONELIFE framework.

**Questions:**

See Weaknesses.

---

> ### Author Response · Authors · 2025-11-21
> **Response to CLdf (Part 1)**
>
> Thank you for the thoughtful review recognizing Crafter-OO as a “**valuable contribution to the research community**” and acknowledging our work “**tackles a challenging and important problem setting**”! We appreciate your helpful feedback regarding the technical positioning and environment details. In response, we have clarified the manuscript by including **a new table contrasting our work with prior work** and **providing the requested specifics on the action space and state representation**.
>
> ## W1: Challenges in Obtaining Symbolic State Observations
>
> > The setup, while realistic in its limited interaction budget, makes several unrealistic assumptions. It relies on full, symbolic observability of a structured state, which is rare.
>
> We agree that full, symbolic observability of a structured state can be difficult to obtain, which further underscores one of the additional key contributions of our work (i.e the Crafter-OO environment). Prior work has mostly focused on environments where such a state can be easily extracted (e.g. visual parsing of arcade environments). For more complex environments, this symbolic, structured state is not available nor easily extracted, which limits the community from *even characterizing the effectiveness of programmatic world modeling methods in more complex environments.* By developing the Crafter-OO environment, we provide the first testbed for work on programmatic world modeling in more complex, stochastic environments where one can focus on the program synthesis and exploration aspects of the programmatic world modeling problem. Our evaluation (Table 1\) shows that even given a fully observable, structured symbolic state, programmatic world modeling is very challenging. We leave state extraction / parsing to future work.
>
> ## W2: Importance of the “One Life” Constraint
>
> > "one life" constraint also feels artificial for many real-world tasks (e.g., robot manipulation) where mistakes are often recoverable
>
> The "one-life" constraint is crucial for modeling high-stakes real-world scenarios where failures are extremely costly or non-recoverable, such as autonomous search-and-rescue in a post-disaster environment or reverse-engineering delicate hardware. This setting motivates the need for rapid, autonomous adaptation to encounters in an unknown environment, which is the focus of our work and a key challenge explored in single-life reinforcement learning \[1\].
>
> We believe working in such scenarios where some mistakes are fatal and may not be recoverable is valuable, as they are often high-stakes and common. In our setting, a fatal, unrecoverable mistake is letting health drop to 0\. In broader Safe RL literature, such fatal mistakes correspond to violating stability guarantees or leaving a safe region of attraction \[2\], breaching hard safety constraints in autonomous driving \[3, 4\], or triggering undesirable behaviors in high-stakes medical applications \[5\].
>
> \[1\] Chen, Annie, Archit Sharma, Sergey Levine, and Chelsea Finn. "You only live once: Single-life reinforcement learning." *Advances in Neural Information Processing Systems* 35 (2022): 14784-14797.
>
> \[2\] Berkenkamp, Felix, Matteo Turchetta, Angela Schoellig, and Andreas Krause. "Safe model-based reinforcement learning with stability guarantees." Advances in neural information processing systems 30 (2017).
>
> \[3\] Shalev-Shwartz, Shai, Shaked Shammah, and Amnon Shashua. "Safe, multi-agent, reinforcement learning for autonomous driving." arXiv preprint arXiv:1610.03295 (2016).
>
> \[4\] Achiam, Joshua, et al. "Constrained policy optimization." International Conference on Machine Learning. PMLR, 2017\.
>
> \[5\] Thomas, Philip S., et al. "Preventing undesirable behavior of intelligent machines." Science 366.6468 (2019): 999-1004.

---

> ### Author Response · Authors · 2025-11-21
> **Response to CLdf (Part 2)**
>
> ## W3: Novel Formalism \+ Inference/Optimization Algorithms for Programmatic World Modeling
>
> > The paper could do a better job of positioning its technical novelty against this prior work, beyond just the Crafter-OO and "one life" setting.
>
> The core technical novelty lies in a new neurosymbolic world model formalism, a novel inference algorithm, and a custom optimization procedure for learning the neural weights. We’ve produced  a table below that describes our positioning against prior work and **lines 176-181 in the related work on domain inference and state tracking**. We develop a novel representational formalism for programmatic world models and develop a novel inference algorithm to sample from the world model, as well as an optimization procedure for fitting the world model.
>
> | Feature | PoE-World | OneLife (Ours) |
> | :---- | :---- | :---- |
> | **Formalism** | **Superposition of Experts:** Each expert predicts the *entire* next state distribution. | **Mixture of Atomic Laws:** Each law predicts a *minimal subset* of attributes (e.g., just NPC position, just movement, etc). |
> | **Inference** | **Static Graph:** All experts contribute to the posterior of all attributes, introducing noise from irrelevant experts. | **Dynamic Graph:** Gradients are routed *only* to laws with satisfied preconditions relevant to the specific transition. |
> | **State Scope** | **Physics-centric:** Position, velocity, health only. | **Arbitrary Attributes:** Inventory, map tiles, NPC states, discrete properties, physics attributes. |
> | **Learning** | **Guided:** Uses human-provided environment rewards/goals. | **Unguided:** Purely observational reverse-engineering. |
>
> Conceptually, the main differences between our representation \+ inference algorithm and PoE-World’s are as follows:
>
> **Formalism** We learn a mixture of *atomic* *laws* that each individually predict a *minimal subset of the next state*, whereas in PoE-World, each of their programmatic experts produces a prediction for the *entire next state*.
>
> 1. Thus, we aim to *factorize the transition function into atomic, predictive laws*, whereas PoE-World effectively learns a *superposition of world modeling expert programs.*
> 2. this causes attribute predictions from PoE-World for complex states to tend towards randomness due to contributions from irrelevant experts.
>
> **Inference** Our optimization procedure creates a dynamic computational graph to route gradients from predictions only to laws relevant to a specific transition.
>
> 1. This is done by taking advantage of the precondition-effect structure of our laws.
> 2. In contrast, PoE-World uses a static computational graph that requires every world modeling expert program to make predictions about every attribute of the world, even ones they do not predict well.
> 3. This causes the posterior of PoE-World’s world model to be very noisy in a complex environment like Crafter-OO.
>
> **State Scope** Our world model is capable of making predictions about arbitrary attributes of the world, such as NPC attributes, tiles in the game map, or items in a player’s inventory.
>
> 1. PoE-World focuses only on modeling a few selected physics-based attributes such as position, velocity, and health.
> 2. This allows our inferred laws to model a diverse range of game mechanics, from movement to crafting to NPC behavior as shown in Appendix B.
>
> ## W4: Rationale for Object-Oriented State Representation \+ Environment Details
>
> > The paper is unclear about the environment's specifics. For instance, it's not immediately clear why the object-oriented state isn't just described as a PDDL-based representation. A clear list or description of the agent's action space is also missing, making it hard to grasp the task's complexity.
>
> We describe the object-oriented state in Python / JSON because we found that it is substantially easier for LLMs to write programs manipulating Python objects or JSON than to write PDDL. Additionally, when described as PDDL, the state becomes very large and substantially increases the time and cost of experiments. Another obstacle to using PDDL are aspects of the environment like probabilistic dynamics. While Probabilistic PDDL can handle this, it makes an already challenging program synthesis task even more difficult for LLMs, since Probabilistic PDDL is likely much rarer in the data than PDDL, which is in turn rarer than Python.
>
> **We’ve explained this rationale in footnote (L215) and L176-181.**

---

> ### Author Response · Authors · 2025-11-21
> **Response to CLdf (Part 3)**
>
> ## W5: Clarifying Engineering Scale and Rigor of the Crafter-OO Environment
>
> > The framework is only evaluated on Crafter-OO. While this is a new and complex environment, testing on only one domain makes it difficult to assess the generalizability of the ONELIFE framework.
>
> Thank you for acknowledging the complexity of the Crafter-OO environment\! We do only evaluate on Crafter-OO, but it features a wide range of mechanics, is stochastic, and significant manual effort went into designing a rigorous evaluation. Additionally, implementing Crafter-OO was a multi-month engineering effort. We do not think it would be possible to add another environment of equal complexity and an evaluation of equal rigor within the timeframe of the rebuttal. We provide a link (https://anonymous.4open.science/r/crafter-oo-iclr26-anon-35C2) to the anonymous version of the environment’s source code — it totals over 4426 lines of code over the original Crafter.
>
> We note that other reviewers (SHXj) state that “Experiments are performed in a single domain. I do not consider this to be a critical flaw…”, We argue that evaluation on Crafter-OO environment provides a more rigorous and complete test than a battery of simpler, limited domains. Crafter-OO is a single, unified environment that simultaneously exhibits the key challenges of our research questions (stochastic dynamics, large number of discoverable mechanics, and high complexity). As reviewer (SHXj) also noted, we believe our contribution to the community, the Crafter-OO testbed itself, is a major step toward future work.

---

> ### Author Response · Authors · 2025-11-26
> **Follow up to reviewer CLdf**
>
> Hi CLdf\! With only a few days left in the discussion period, we wanted to gently check in whether our rebuttal addressed all your questions. We are happy to address any remaining questions. **We’ve made several revisions to the manuscript to address issues you identified**. We hope that these additional results and answers will allow you to revisit your score — otherwise, we are happy to engage further\!

---

### Author Response · Authors · 2025-12-02
**Discussion Summary for Area Chair**

Thanks AC + reviewers! This note summarizes the reviewer consensus and the improvements made during the rebuttal period. We addressed all questions on baselines, long-term planning, and positioning via new experiments and PDF revisions.

#### **Reviewer Consensus**

* “tackles a challenging and important problem setting” (CLdf)
* “valuable contribution to the research community” (CLdf)
* “I consider significance to be one of the strengths of the paper” (SHXj)
* “firmly within an acceptable level of quality for publication” (SHXj)
* “New environment Crafter-OO is interesting” (NpK1)
* “good to see that symbolic world models also work in the Crafter environment”  (NpK1)

Before the freeze, our scores were 8 (SHXj), 6 (NpK1), and 4 (CLdf). CLdf had not responded to our rebuttal yet.

### **Rebuttal Improvements**

#### **Additions for Reviewer SHXj**

***(Score raised from 4 to 8 on Nov 25\)***

**Experiments and Analysis**

* **Added WorldCoder Baseline:** We integrated WorldCoder (Tang et al., 2024\) as a second baseline. OneLife outperforms WorldCoder in both Rank@1 (18.7% vs 0.0%) and State Fidelity.
* **Visualized Rank Distribution:** We added histograms and violin plots (Appendix J) to display the raw rank distribution of ground truth states. This clarifies the model's discriminative performance beyond the Mean Reciprocal Rank metric.
* **Weight Ablation Study:** We added an ablation study to Table 1 demonstrating that removing the learnable weights reduces Rank@1 performance by 5.7%. This justifies the inclusion of the learnable weights.
* **Multi-step Planning Evaluation:** (Also requested by NpK1) We demonstrated the model's ability to rank effective plans over ineffective ones in multi-step scenarios (Appendix A).

**Writing and Clarification**

* **Pure vs. Probabilistic Functions:** We added Appendix H to explain why we model a "pure" transition function probabilistically. We argue that a deterministic model would require overfitting to the simulator's specific PRNG scheduling, whereas a probabilistic model captures robust macroscopic laws (L196-198, L2168-2215).
* **Exploration Policy Domain Knowledge:** We clarified that the exploration policy uses general genre priors (e.g., "hostile creatures exist") but not environment-specific dynamics (e.g., "zombies chase players"), mimicking a realistic reverse-engineering setting (L320-327).
* **Model Scope:** We clarified that OneLife synthesizes a single, compositional world model from the unguided exploration phase, which is then evaluated across the diverse scenarios, rather than training separate models per scenario.
* **Space of Laws:** We added a precise definition of the space of synthesized laws, specifying that they cover categorical and discrete distributions over state attributes (L275-277).
* **Synthesizer Implementation:** We reworked the description of the synthesizer (Section 3.3) to explicitly detail the change detection routine and LLM generation process.
* **Mathematical Notation:** We revised the notation in Section 3.2 to rigorously define the mapping to distributions over next-state attributes.

#### **Additions for Reviewer NpK1**

**Experiments and Analysis**

* **New Planning Evaluation:** We added a planning section (Appendix A). We tested the world model on three multi-step scenarios: Zombie Fighter, Stone Miner, and Sword Maker. The world model correctly ranked effective plans higher than ineffective ones in all cases. This demonstrates the model captures causal dynamics necessary for decision making.

**Writing and Clarification**

* **Synthesizer Differentiation:** We clarified the distinction between our approach and PoE-World. PoE-World relies on 30+ hand-crafted, mechanic-specific synthesizers. OneLife uses a single general-purpose synthesizer that identifies mechanics from raw state changes.

#### **Additions for Reviewer CLdf**

**Writing and Clarification**

* **Comparison with Prior Work:** We added a table contrasting OneLife with PoE-World. We highlighted that OneLife factorizes dynamics into atomic laws with a dynamic computation graph. PoE-World uses a static graph where all experts contribute to all predictions.
* **Rationale for Object-Oriented State:** We explained why we use a Python/JSON state representation instead of PDDL (L215, L176-181). We noted that PDDL representations for complex environments become prohibitively large and that LLMs synthesize Python more effectively than Probabilistic PDDL.
* **"One Life" Constraint:** We expanded the motivation to connect the "one life" constraint to Safe RL contexts where failures are non-recoverable.
* **Environment Details:** We provided the requested formal description of the action space and state representation.
* **Code Release:** We provided [an anonymous repository](https://anonymous.4open.science/r/crafter-oo-iclr26-anon-35C2/README.md) containing the full source code for the Crafter-OO environment (4426+ lines of code).

---

### Meta-Review · Area_Chair_ZXoa · 2026-01-11

**Summary:**

All reviewers appreciated the authors’ contribution to the challenging problem of learning a symbolic world model in a complex environment with severe constraints, and the main concerns were about unrealistic assumptions, unclear technical novelty, missing analyses, and limited experiments. The authors provided a thorough rebuttal, clarifying missing details and conducting additional experiments/analyses with more baselines. In particular, they provided an additional benchmark result on three multi-step scenarios, Zombie Fighter, Stone Miner, Sword Maker, demonstrating that the proposed model can capture causal dynamics necessary for decision making. After discussion, reviewer SHXj raised his/her original rating of 4 to 8, acknowledging the acceptable level of quality. Despite some unrealistic limitations in the experimental setup, AC also finds that this work is an interesting contribution to world modeling in a challenging environment and thus recommends acceptance. AC strongly encourages the authors to carefully incorporate the reviewers’ comments and the rebuttal content into the final manuscript.

**Reviewer Concerns:**

The main concerns were about unrealistic assumptions, unclear technical novelty, missing analyses, and limited experiments. The authors provided a thorough rebuttal, clarifying missing details and conducting additional experiments/analyses with more baselines.
For example, regarding novelty over prior work, i.e., PoE-World, they clarified that their method uses a single general-purpose synthesizer to identify mechanics from raw state changes, whereas PoE-World relies on 30+ hand-crafted, mechanic-specific synthesizers.  In particular, they provided an additional benchmark result on three multi-step scenarios, Zombie Fighter, Stone Miner, Sword Maker, demonstrating that the proposed model can capture causal dynamics necessary for decision making. After discussion, reviewer SHXj raised his/her original rating of 4 to 8, acknowledging the acceptable level of quality.

**Reviewer Scores:**

Reviewer CLdf would have changed the score from 4 to 6.
Reviewer NpK1 would have changed the score from 6 to 8.
Reviewer SHXj changed the original rating of 4 to 8 after discussion.

---

### Decision · Program_Chairs · 2026-01-26

Accept (Poster)